# Magnetic vector tomography reveals giant magnetofossils are optimised for magnetointensity reception
Richard J. Harrison [1,12] ✉, Jeffrey Neethirajan [2,12], Zhaowen Pei [3,4], Pengfei Xue [5], Lourdes Marcano [6,7], Radu Abrudan [8], Emilie Ringe [1], Po-Yen Tung [1], Venkata S. C. Kuppili [9], Burkhard Kaulich [10], Benedikt J. Daurer [10], Luis Carlos Colocho Hurtarte [10], Majid Kazemian [10], Liao Chang [3] ✉, Claire Donnelly [2,11] ✉ & Sergio Valencia [8] ✉

Giant magnetofossils are unusual, micron-sized biogenic magnetite particles found in sediments dating back at least 97 million years. Their distinctive morphologies are the product of biologically controlled mineralisation, yet the identity of their biomineralising organism, and the biological function they serve, remain a mystery. It is currently thought that the organism exploited magnetite's mechanical properties for protection. Here we explore an alternative hypothesis, that it exploited magnetite's magnetic properties for the purpose of magnetoreception. We present a three-dimensional magnetic vector tomography study of a giant magnetofossil and assess its magnetoreceptive potential. Our results reveal a single magnetic vortex that displays an optimised response to spatial variations in the intensity of Earth's magnetic field. This magnetic trait may have conferred an evolutionary advantage to mobile marine organisms, providing an upper age limit on the development of navigational magnetoreception and raising the possibility that earlier evidence of this sense may yet be preserved in the fossil record. More broadly, this work provides a blueprint for assessing the morphological and magnetic evidence for putative biogenic iron oxide particles, which are a key component in the search for early life on Earth and Mars.

The 2008 discovery of putative giant magnetofossils in marine sediments from a period of intense global warming ~56 million years ago sparked debate surrounding the biological mechanisms responsible for their formation, the ecological and evolutionary contexts that favoured their development, and the biological functions they might have served[1–3]. Giant magnetofossils were initially thought to be exclusively associated with hyperthermal events, and therefore considered useful indicators of past climates and environments. However, they have since been identified globally in sediments both modern and ancient and from periods of global cooling as well as warming[4–11]. Their widespread occurrence[6], chemical purity[1], oxygen isotopes[1], crystallographic perfection[3], crystallographic

orientations[12], allometric relations[1,5] and distinctive morphologies (comprising needles, spindles, giant bullets, kinked giant bullets, socks and spearheads[12], and potentially also seeds, squash and spades[4]) provide compelling evidence of biogenicity, although until the organism responsible for making them is identified, this evidence remains circumstantial. The characteristics of giant bullet-shaped magnetofossils, for example, are identical to those of conventional bullet-shaped magnetofossils created by magnetotactic bacteria in all respects other than their overall size (1–2 μm for giant bullets versus 100–200 nm for conventional bullets) and crystallographic orientation (length parallel to <111> for giant bullets versus <100> and more rarely <110> for conventional bullets[12–15]). Due to their large size,

[1]Department of Earth Sciences, University of Cambridge, Cambridge, UK. [2]Max Planck Institute for Chemical Physics of Solids, Dresden, Germany. [3]SKLab-DeepMinE, MOEKLab-OBCE, School of Earth and Space Sciences, Peking University, Beijing, China. [4]State Key Laboratory of Coal Mine Disaster Dynamics and Control, Chongqing University, Chongqing, China. [5]State Key Laboratory of Marine Geology, Tongji University, Shanghai, China. [6]Department of Physics and Nanomateriales Magnéticos para Energía y Salud (MAGNES), University of Oviedo, Oviedo, Spain. [7]Center for Cooperative Research in Biomaterials (CIC biomaGUNE), Basque Research and Technology Alliance, Donostia-San Sebastián, Spain. [8]Helmholtz-Zentrum Berlin für Materialien und Energie GmbH, Berlin, Germany. [9]Canadian Light Source, Saskatoon, Canada. [10]Diamond Light Source, Didcot, UK. [11]International Institute for Sustainability with Knotted Chiral Meta Matter (WPI-SKCM2), Hiroshima University, Hiroshima, Japan. [12]These authors contributed equally: Richard J. Harrison, Jeffrey Neethirajan. ✉e-mail: rjh40@cam.ac.uk; liao.chang@pku.edu.cn; Claire.Donnelly@cpfs.mpg.de; sergio.valencia@helmholtz-berlin.de

giant magnetofossils are not considered to be the product of bacteria, but rather (as yet unidentified) eukaryotes[1]. The most common interpretation is that the biomineralising organism exploited the mechanical hardness of magnetite for protection from predation[1]. Evidence supporting this hypothesis comes in the form of radiating clusters of up to $100 \sim 3\,\mu m$ spearhead-like crystals, likely in an intact orientation (tip outwards) from the original organism[1]. These clusters are proposed to be spicules forming the abrasive armament on the skin of some worm-like creature in the mud, where shear forces are able to overcome the magnetic forces holding the cluster together, producing some of the isolated crystals that are more typically observed. Iron-rich dermal spicules are well known in biology and can be found disaggregated in sediments[16–18]. Mesocrystalline magnetite is also known to be biomineralised in the major lateral teeth of chitons[19,20], where it is used as a hardening agent to scrape endolithic algae from rocks. Clusters of giant magnetofossils have not been reported in subsequent studies, however, and none containing other types of giant magnetofossil (e.g., giant bullets, needles, spindles, etc.) are known. An alternative hypothesis, considering their morphological similarity to some conventional magnetofossils, is that these crystals were exploited to perform some kind of biomagnetic function[3,5,6,9,10]. Magnetotactic bacteria synthesise intracellular magnetic nanoparticles with close biological control of their size, shape, crystallographic orientation and spatial arrangement. This control yields chains of uniformly magnetised particles that are highly optimised for the purpose of magnetotaxis[21,22] (i.e., the physical rotation of the bacterial cell into parallel alignment with the Earth's magnetic field, enabling it to perform an efficient, one-dimensional search for its optimal position in a chemically stratified environment). In contrast, we know little about the magnetic structure of giant magnetofossils, and, crucially, whether they display any evidence that their magnetic properties are optimised to perform biomagnetic functions such as magnetointensity reception[23–26] (i.e., the ability of an animal to sense the intensity of the Earth's magnetic field via the torque exerted on magnetite particles inside specialised receptor cells). Finding such evidence would add further support for a biogenic origin of giant magnetofossils. Micromagnetic simulations predict a range of possible magnetic structures in giant magnetofossils, from uniformly magnetised needles to non-uniformly magnetised spindles, giant bullets and spearheads[3,5,9]. However, these predicted states have yet to be confirmed experimentally and non-uniform magnetic states (particularly the complex multi-domain states predicted for spearheads) are usually considered poorly optimised from a magnetic sensing perspective. Here we reconstruct the three-dimensional (3D) internal magnetic structure of a giant spearhead magnetofossil using magnetic vector tomography[27,28]. Armed with this knowledge, we calculate its magnetoreceptive response using a torque-transducer model[29] and demonstrate its optimised potential to sense the Earth's magnetic field intensity.

## Results
### Magnetic vector tomography
The sample studied is a "no-stalk" spearhead extracted from a $\sim 56$ million year old pelagic marine sediment from the J-Anomaly Ridge, North Atlantic[5] (Fig. 1a). The particle has a diameter of 1.1 μm and a length of 2.25 μm, and comprises an approximately cylindrical base and a cone-shaped tip (Fig. 1b). The dimensions of the particle make it highly absorbing to resonant soft X-rays and electrons, which poses a major challenge to probing its internal magnetic structure without destructive sampling (which would irreversibly change its magnetic state). Most transmission-based nanomagnetic imaging methods, such as electron holography[30] and scanning transmission X-ray microscopy[31], are limited to samples thinner than ~300 nm. Recent technical breakthroughs, however, open up the opportunity to image natural Fe-oxide samples in the multi-micron size range[27,28,32]. Soft X-ray pre-edge dichroic ptychography[32] works by tuning soft X-rays to energies just below the Fe-absorption edge, enabling them to pass through much larger samples. Magnetic vector tomography[27,28] then enables all three components of magnetisation to be reconstructed and spatially resolved throughout the

volume of the grain with a resolution of a few tens of nm. This combination of techniques opens up the entire single- to multi-vortex size range of natural remanence carriers to 3D magnetic imaging. Crucially, these breakthroughs mean that the magnetic information accessible to experimental observation matches precisely what can be accessed through micromagnetic simulation. It is now possible, therefore, to test and verify the predictions of micromagnetic theory by direct comparison of predicted versus observed behaviour in 3D at the individual grain scale – a capability that will impact the broader fields of rock magnetism, paleomagnetism and environmental magnetism.

Here we probe the magnetic configuration of the spearhead magnetofossil non-destructively using pre-edge phase X-ray magnetic circular dichroism (XMCD)[32] (see Methods). High-resolution projections of the sample were obtained using ptychography[33,34], which yields both the amplitude and phase of the reconstructed image. Images were acquired using left and right circular polarised radiation and a photon energy within the Fe $L_3$ pre-edge region (top panels Fig. 1c–h). A series of 2D phase XMCD images taken at different rotation angles about two perpendicular rotation axes highlight key aspects of the internal magnetic domain structure (bottom panels Fig. 1c–h). Whenever the long axis of the particle ($z$) is oriented normal to the X-ray beam (Fig. 1d, Fig. 1f–h), strong positive and negative magnetic contrast is observed in the left and right lateral halves throughout the entire length of the particle, indicating that magnetisation predominantly lies in the plane perpendicular to the symmetry axis of the spearhead. The white line separating regions of opposing magnetisation displays a curved/kinked trajectory. When $z$ is strongly tilted with respect to the X-ray beam direction (Fig. 1c, e), a strengthening or weakening of magnetic contrast in the body, and a complimentary weakening or strengthening of contrast in the tip of the particle, is observed. This observation reveals that there is an additional component of magnetisation along the particle length, and that this length-parallel component is different in the body versus the tip of the particle.

To determine the 3D magnetic configuration of the sample, magnetic vector tomography was performed using phase XMCD projections acquired about two rotation axes and the 3D magnetisation vector field reconstructed with a dedicated algorithm[27,28] (Fig. 2, Fig. S1, Supplemental Movies S1 and S2). The dominant feature is a single vortex with curved/kinked core trajectory (Fig. 2a), rather than the multi-domain state previously predicted for slightly larger spearheads with a stalk[3,5] or the uniform state previously claimed on the basis of 2D electron holography observations[1]. Taking a cross section through the particle, the magnetisation can be seen to smoothly rotate in the plane, forming a vortex-like texture (Supplemental Movie S1). In the centre of the vortex, the magnetisation is forced to point out of the plane, forming a vortex core, with an associated direction, known as the polarisation. When the position and polarisation of the vortex core are tracked through the volume of the particle, the core is seen to follow a 2D path within the medial plane, initiating at the base of the particle at one edge, moving to the centre of the particle, and then exiting the particle close to the tip. Although the sense of circulation remains constant throughout the particle, when we follow the polarisation of the vortex, the core is seen to reverse abruptly in the centre of the particle (i.e., the change from strong red to strong blue colour in Fig. 2a). Such a core reversal represents a change in topology of the vortex and is mediated by the presence of a Bloch point singularity - a topological defect at the centre of which the magnetisation locally vanishes (Fig. 2a inset). Such Bloch point singularities have been observed in fabricated samples - both bulk, thin film, and nanostructured arrays[27,35–37]. However, this is to the best of our knowledge the first observation of a Bloch point singularity in a natural sample. The curved core trajectory lies close to a medial domain wall, which separates lateral halves of the particle with opposing length-parallel ($M_z$) components of magnetisation (Fig. 2h). A reversal of $M_z$ components occurs in the tip of the particle (Fig. S1), which can be visualised as a twisting of the medial domain wall (Fig. 2j). The kink in the

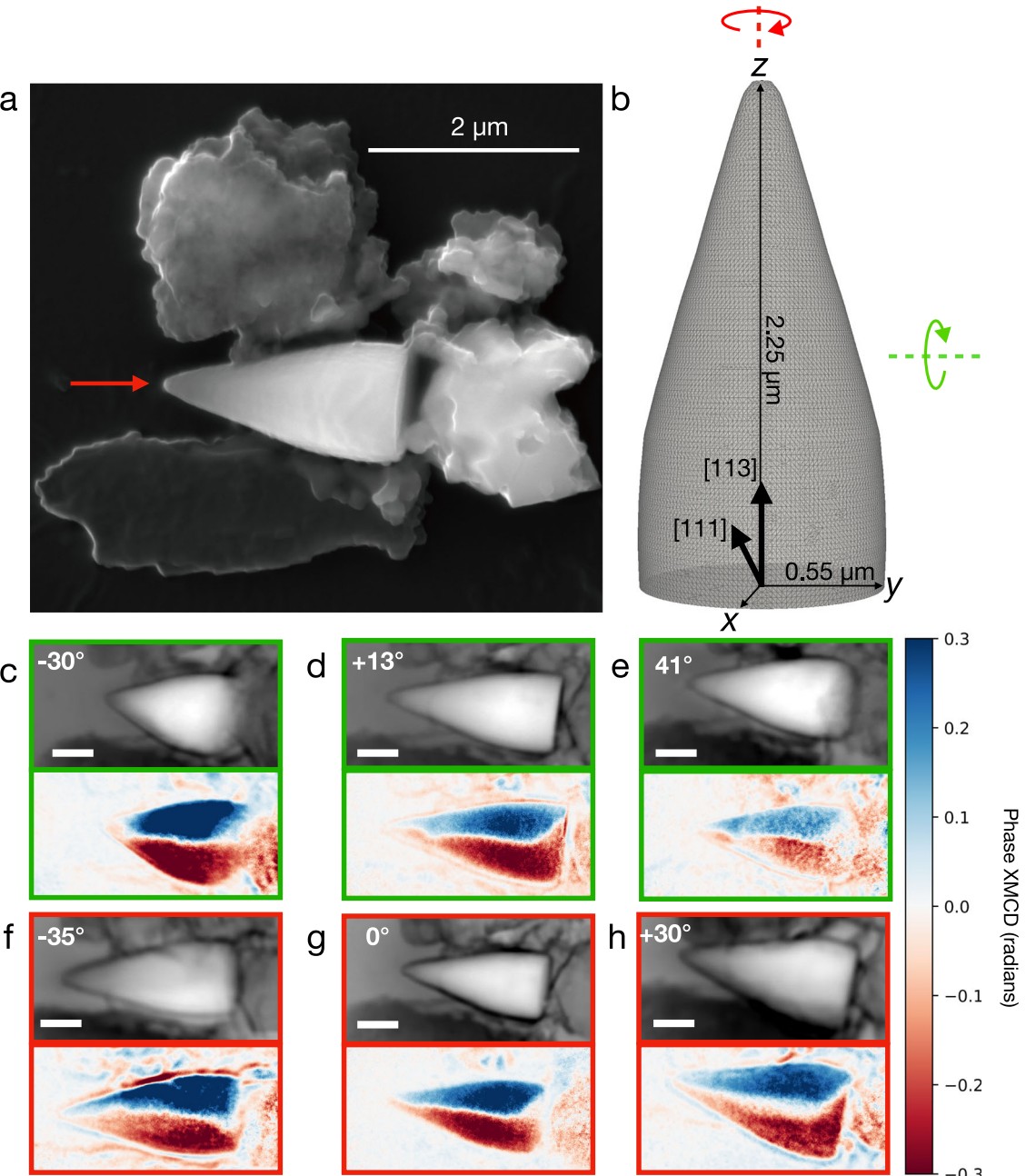

**Fig. 1 | 2D microscopic and magnetic characterisation of a giant spearhead magnetofossil. a** Scanning electron microscopy image of the 'no-stalk' spearhead giant magnetofossil (red arrow) surrounded by detrital particles. **b** Example 3D model of the particle used in micromagnetic simulations. **c–h** 2D projections of the phase (upper panel) and the phase XMCD signal (lower panel) obtained using pre-edge dichroic ptychography[32]. The X-ray beam is normal to the plane of the images

(along $x$) and has an energy of 706 eV, 4.5 eV below the observed peak of the absorption edge. For the magnetic images, blue and red colours correspond to magnetisation projections into and out of the plane of the diagram, respectively. 2D projections are shown for three rotation angles about axes approximately parallel to $y$ (**c–e**) and $z$ (**f–h**). Scale bar in (**c–h**) 500 nm.

core trajectory is associated with the interplay between the core and the $M_z$ reversal in the tip (Fig. S1).

## Micromagnetic modelling

All key features of the observed 3D magnetic topology can be reproduced in micromagnetic simulations[38] (Fig. 3, Figs. S2–5, Supplemental Movies S3–6). The lowest energy state consists of a curved single vortex core and medial domain wall, in excellent agreement with the experimental observations, but lacking Bloch point or tip domain. Placing the magnetite [113] direction parallel to $z$ provided the best fit to the observations, in

agreement with expectations based on electron diffraction measurements of other "no-stalk" spearhead giant magnetofossils (Fig. S2, Table S1). The same low-energy state was obtained by modelling growth of the particle in 25 nm steps from either the tip down or the base up[14,39] (Supplemental Movies S4 and S5). A Bloch point can be nucleated at the tip by application of a magnetic field on the order of 10–20 mT (Fig. S3). The Bloch point propagates along the length and exits at the base of the particle, enabling core reversal. The energy barrier for Bloch point nucleation is very high (Fig. S4; Supplemental Movie S6), indicating that it is unlikely to arise spontaneously through thermal activation. These results suggest that the Bloch point and

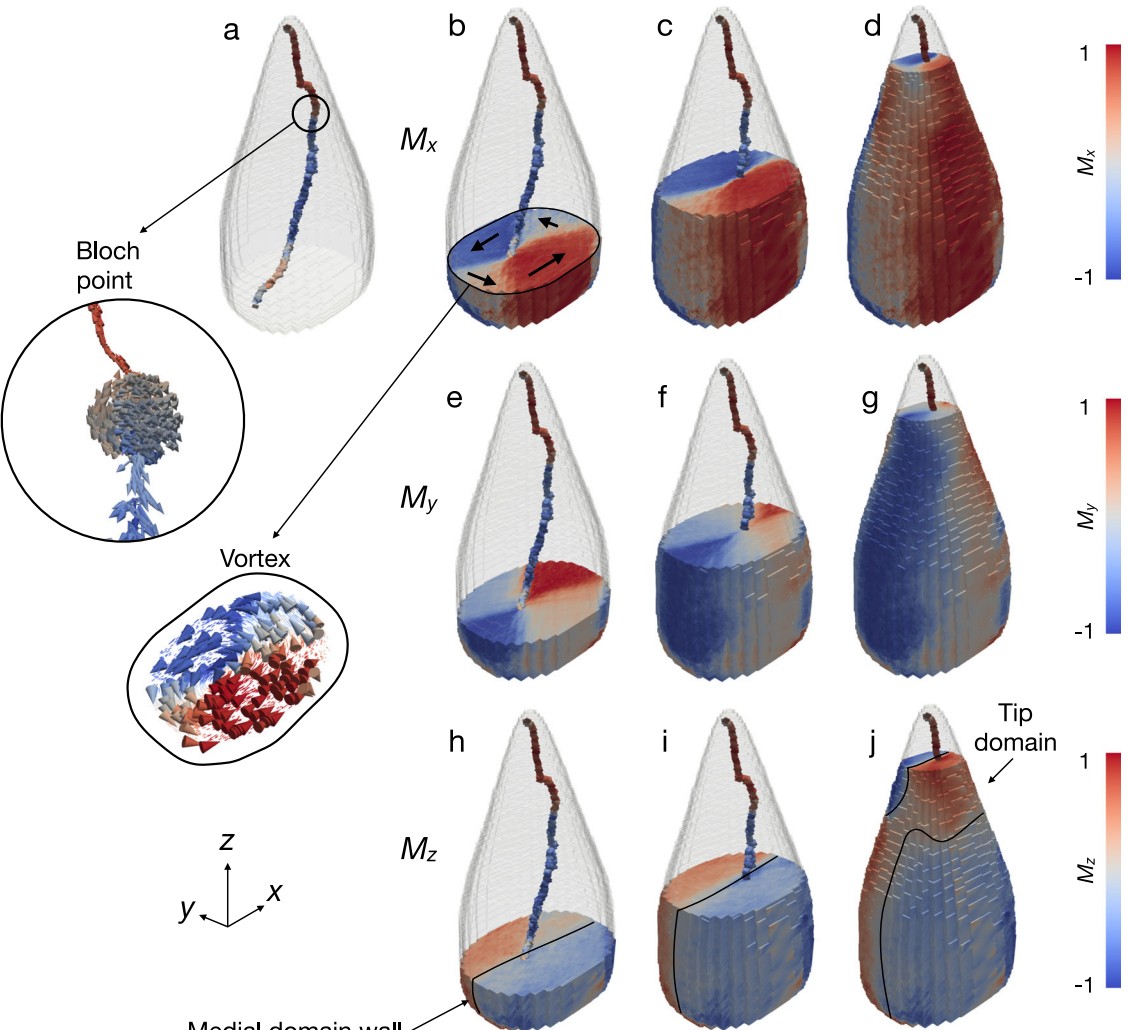

**Fig. 2 | 3D magnetic vector tomography of the giant spearhead magnetofossil.**
**a** Trajectory of the vortex core, coloured according to $M_z$ (red positive, blue negative). Circled region highlights the Bloch point, where the vortex core moment changes sign. An expanded view of the magnetisation vector field surrounding the Bloch point is shown inside the circle. **b**–**d** $M_x$, (**e**–**g**) $M_y$ and (**h**-**j**) $M_z$ components of magnetisation shown at three selected positions along the fossil length. Black arrows in (**b**) illustrate the vortex structure in the base of the particle (see also Supplemental Movie S1), and a slice through the magnetisation vector field at this position is shown in the expanded region. Solid line in (**h**) shows the position of the medial domain wall separating regions with opposing $M_z$. The $M_z$ components in the tip of the particle are reversed with respect to the base, leading to the twisting of the medial wall in (**j**) (see also Fig. S1).

tip domain are likely caused by exposure of the sample to a magnetic field during magnetic extraction from the sediment (see Methods), rather than being primary features of the fossil's natural, lowest energy magnetic state.

The excellent agreement between observed (Fig. 2) and simulated (Fig. 3) 3D magnetic structures provides confidence in our use of micromagnetic simulations to predict the magnetic properties of the giant spearhead. The remanent magnetic moment, which is dominated by the vortex core, is $1 \times 10^{-14}$ Am$^2$ oriented at ~10° to $z$. The stability of the remanent moment is high (from a biological perspective), with coercivity $B_c$ ~ 2 mT for fields applied along $z$ (40 times greater than the Earth's field) and coercivity of remanence $B_{cr}$ ~ 12 mT (240 times greater than the Earth's field) (Fig. 3f, g). Arguably the key property that distinguishes giant magnetofossils from their conventional counterparts is their high susceptibility, which is up to 7500 times greater in terms of induced moment per unit field than typical magnetotactic bacteria ($6.3 \times 10^{-12}$ Am$^2$/T for the giant spearhead versus $8.35 \times 10^{-16}$ – $1.5 \times 10^{-15}$ Am$^2$/T for magnetotactic bacteria[40], see Supplemental Information). The susceptibility is anisotropic, with induced moments along $z$ a factor of 2.5 times higher than along $x$ or $y$ (see Supplemental Information). Whereas magnetotactic bacteria display square hysteresis loops

for fields applied along the chain (i.e., the squareness ratio of saturation remanence to saturation magnetisation $M_{rs}/M_s$ ~ 1), the giant spearhead shows a loop with much lower squareness ratio for fields applied along $z$ ($M_{rs}/M_s$ ~ 0.02). Induced changes in moment as a function of field are accommodated by lateral movement of the medial domain wall, particularly towards the outer edges of the particle.

In terms of potential non-magnetoreceptive applications, it is worth noting that the observed microstructural and magnetic characteristics of the giant spearhead magnetofossil are very different from those of magnetite produced by other organisms for non-magnetic purposes (e.g., the magnetite nanorod/chitin fibre composite structure produced for hardening/abrasion purposes by chiton[20] or the ultrafine-grained magnetite particles produced as a metabolic byproduct by dissimilatory iron-reducing bacteria[41]). Despite the lower $M_{rs}/M_s$ ratio, the remanent magnetic moment of the giant spearhead studied here is large in absolute terms – equivalent to over 300 uniformly magnetised single-domain particles (assuming a 50 nm diameter) and at least a factor of 1.6–3.4 times larger than the remanent moment of a single spherical or cuboctahedral particle with equivalent volume (Fig. S5). The elongated shape of the fossil also means its remanent moment, originating from the vortex

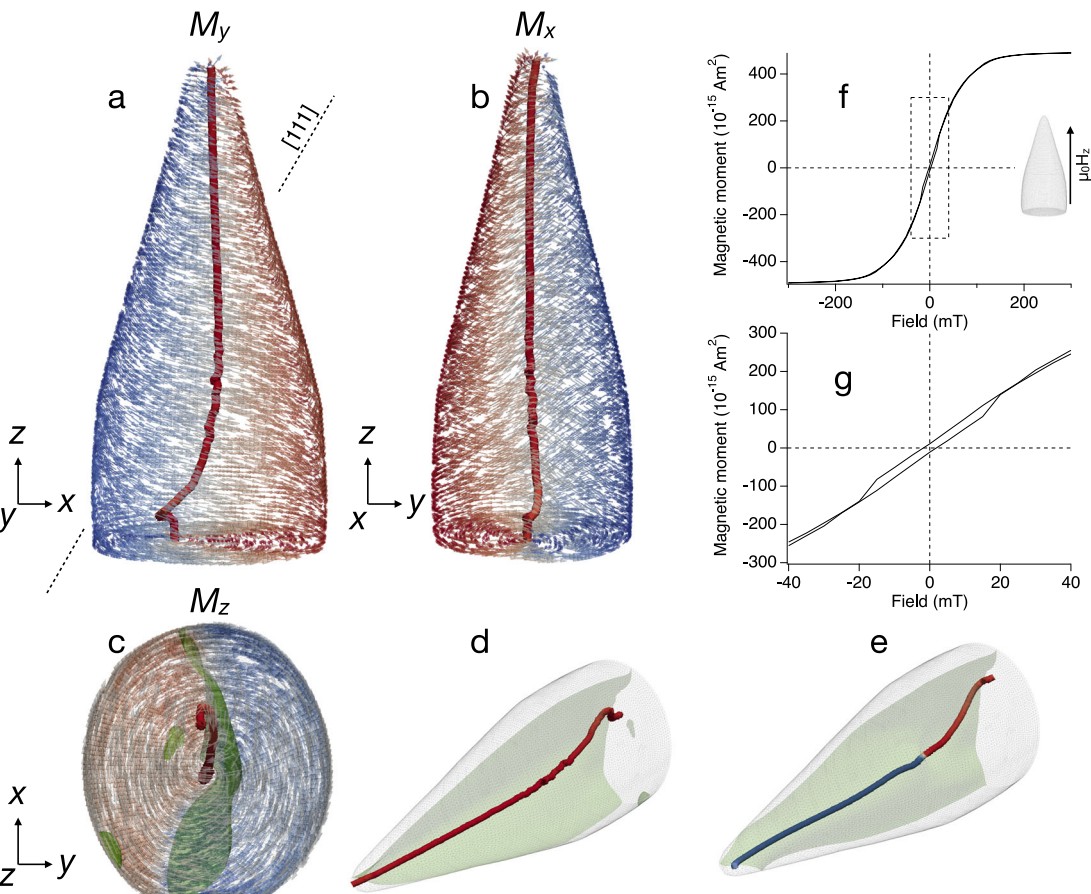

**Fig. 3 | Micromagnetic simulations of a giant spearhead magnetofossil.** Simulations were performed with an elliptical cross section ($r_x = 0.55$ µm; $r_y = 0.4675$ µm), [113] parallel to $z$ and [111] (the magnetocrystalline easy axis) in the $x$-$z$ plane. **a–d** Visualisations of the lowest energy remanence state. Surface spins are coloured according to the component of magnetisation in the corresponding view direction: (**a**) $M_y$, (**b**) $M_x$ and (**c**) $M_z$ (red is positive, blue is negative). The vortex core is coloured according to $M_z$. The green surface in (**c**) and (**d**) represents $M_z = 0$, i.e., the medial domain wall separating lateral halves of the particle with positive and negative $M_z$ components. **e** Visualisation of the remanent state obtained after taking the lowest energy state in (**a**), applying a $-26$ mT back field along $z$, and then switching the field off. The change in colour of the vortex core from red to blue corresponds to a Bloch point (see inset to Fig. 2a). **f** A simulated hysteresis loop in field of $\pm 300$ mT applied parallel to $z$. An expanded view of the region highlighted by the dashed box is shown in (**g**).

core, lies in a predictable direction close to the particle's long axis, rather than along a random choice of the four <111> easy axis directions in an equidimensional particle (Fig. S5). In the context of magnetotaxis, the alignment efficiency can be quantified by the ratio of magnetic to thermal energy ($MB/k_BT$, where $M$ is the remanent magnetic moment in Am$^2$, $B$ is the Earth's field in Tesla, $k_B$ is Boltzmann's constant, and $T$ is temperature). A value of $MB = 121\ k_BT$ is found for the giant spearhead, assuming a representative Earth field strength of 50 µT and a temperature of 298 K. This is more than ten times larger than the typical value of $\sim 10\ k_BT$ for magnetotactic bacteria[24], and, therefore, could be considered overengineered (and an inefficient use of metabolic iron) if used solely for the purpose of magnetotaxis. Large remanent moments have also been observed, however, in multicellular magnetotactic prokaryotes[42,43] and single-celled eukaryotes[44], which biomineralise a large number of magnetic particles. The eukaryotic cells are interesting from the perspective of bridging the gap between the evolution of magnetotaxis in magnetotactic bacteria and the origins of magnetite-based magnetoreception in eukaryotes[45]. There is genetic evidence that eukaryotes co-opted the gene set for magnetic particle formation from prokaryotes[46,47]. Since single-celled eukaryotes are able to biomineralise bullets and adapt their size and shape in a similar way to their giant counterparts (albeit not as extreme)[44], it seems reasonable to propose that there is potential for other single-celled, or even multi-cellular eukaryotes (i.e. animals), to go further down the road of increasing magnetic particle size (rather than

simply increasing their number) if there is an evolutionary advantage to do so in terms of magnetoreception.

## Modelling magnetoreceptive potential

Although the nature and origin of magnetoreception in animals remains a controversial topic[48–51], one of the most compelling proposals for a magnetite-based magnetoreceptor has been identified in the olfactory lamella of trout, which was found to contain a $\sim 1$ µm long chain-like cluster of magnetite nanoparticles with a net moment of $5 \times 10^{-16}$ Am$^2$ ($MB = 6\ k_BT$ in a 50 µT field at 298 K)[52]. A follow up study with modified experimental protocol identified additional candidate magnetite-based receptor cells containing dense clusters of magnetite nanoparticles with overall size 1–2 µm and magnetic moments in the range of $4$–$100 \times 10^{-15}$ Am$^2$ ($MB = 50$–$1200\ k_BT$)[53]. Although these results were later challenged as being due to extracellular contamination[54], that result may itself be due to the failure to remove contamination in the reagents and chemicals used for disaggregating the trout olfactory epithelium[55]. Structures similar to those originally described by ref. 52 have since been confirmed by clean-lab extraction of magnetic cells from the olfactory epithelium of salmon[46]. To tackle the question of what evolutionary advantage might be conveyed by replacing clusters of individual nanoparticles with a single, giant magnetofossil, we first consider whether the combination of size, shape and crystallographic orientation of a giant spearhead imparts magnetic properties that are well optimised from a

**Fig. 4 | Modelling the magnetoreceptive potential of giant magnetofossils. a** Schematic model of the torque-transducer mechanism (modified after[29]). The magnetic particle with remanent and/or induced moment *M* is attached to a torsional elastic pivot (vertical grey bar) with stiffness *K*. The rest position of the particle (grey position) corresponds to no elastic deformation of the pivot. A magnetic field *B* applied at an angle *θ* to the rest position creates a magnetic torque, causing the particle to rotate (blue position). As the particle rotates, the elastic pivot is deformed, creating a restoring elastic torque that balances the magnetic torque at an equilibrium deflection angle *ψ*. Thermal energy causes fluctuations Δ*ψ* of amplitude about the equilibrium deflection angle (red shaded region). A filament (yellow) attached to the base of the particle pulls on a force-gated transmembrane ion channel, causing it to open or close. **b** Plot of Δ*ψ* as a function of *MB/K* for the case of magnetic fields applied antiparallel to the rest position (*θ* = 180°)[29] (Supplemental Information). Lines are shown for two different values of the torsional stiffness: *K* = 6 *k_BT* (dashed line) and *K* = 121 *k_BT* (solid line). The shaded region highlights the critical point around *MB/K* = 1 where Δ*ψ* varies most rapidly as a function of field intensity for a given value of *K*. The inset shows the derivative of Δ*ψ* with respect to *MB/K*. Around the critical point, the sensitivity of Δ*ψ* to small variations in *B* for a fixed value of *K* is ~2.5 times higher for a 'giant-like' particle with *MB* = 121 *k_BT* attached to an elastic pivot with *K* = 121 *k_BT* (solid line) compared to a minimum baseline 'trout-like' cluster[52] with *MB/K* = 6 *k_BT* attached to an elastic pivot with *K* = 6 *k_BT*. **c** Plot of d(Δ*ψ*)/d(*MB/K*) at *MB/K* = 1 for a range of *K* (left axis, black line), the minimum detectable angle of deviation from antiparallel alignment with the magnetic field[56], *η* (right inner axis, blue line), and a histogram of calculated magnetic energies for a range of giant magnetofossil morphologies (far right axis, red shaded bars). Symbols show magnetic energy clusters for giant bullets and needles, spearheads, kinked giant bullets, and spindles.

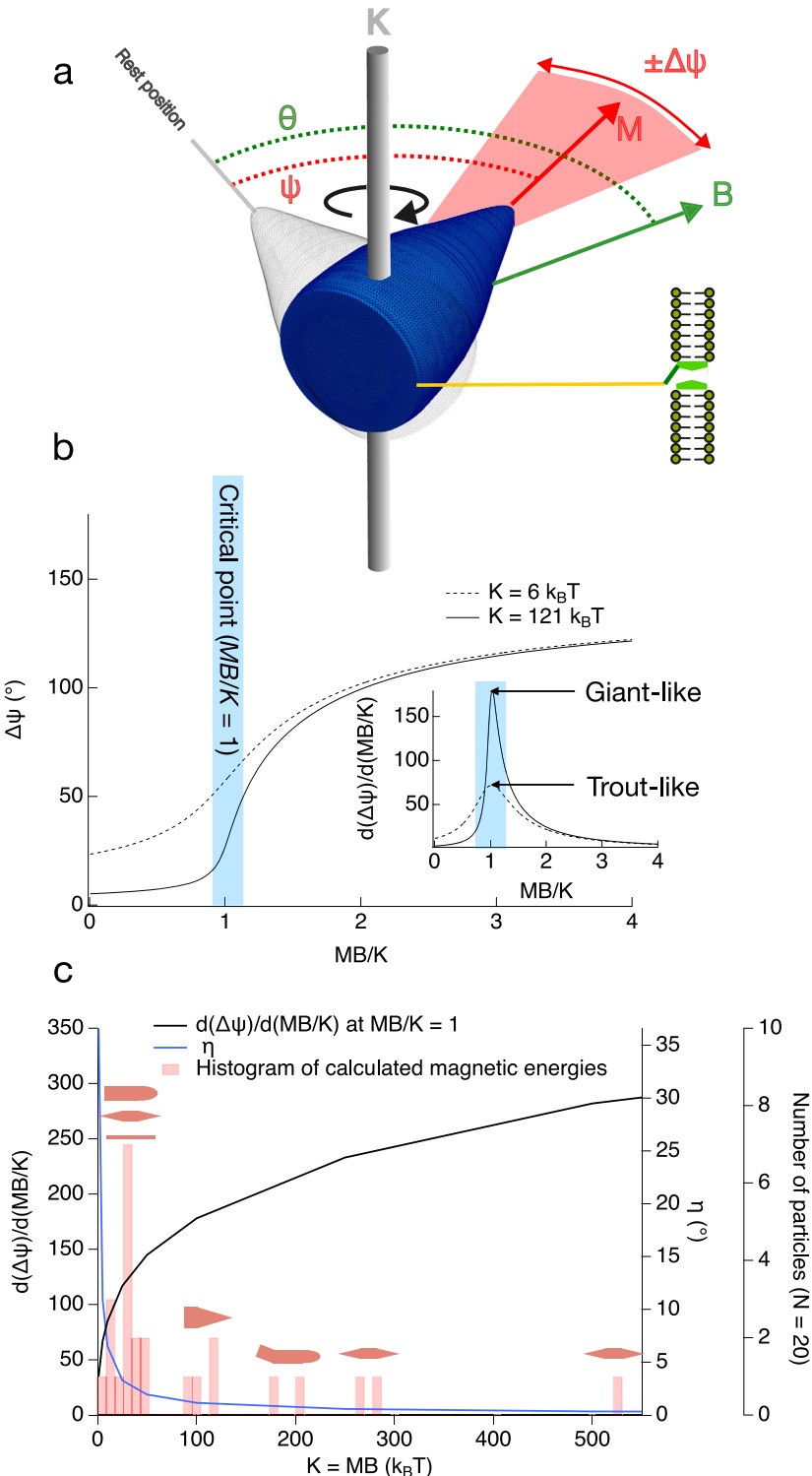

biomagnetic sensing perspective. A useful framework for assessing magnetoreceptive potential is the torque-transducer model[29], which describes the field response of a magnetic particle (or chain of particles) elastically coupled to a magnetoreceptive cell as a thermodynamic balance between magnetic, elastic and thermal energies (Fig. 4). The magnetic energy is created by the Earth's magnetic field acting on the remanent and/or induced magnetic moment of the particle. The elastic energy is created by deformation of an anchoring elastic pivot attached to the particle. Both these energies are comparable to thermal energy, which leads to fluctuations in

moment orientation and elastic deformation. In this model, the deflection characteristics (defined by both the mean deflection angle, *ψ*, and the amplitude of thermal fluctuations around that mean, Δ*ψ*) are a function of the absolute and relative values of magnetic energy (*MB*) and torsional stiffness (*K*), with larger values of *MB* and the ratio *MB/K* (≥ 1) being seen as advantageous in terms of magnetoreception performance (allowing greater deflections, enabling magnetoreception during periods of weaker field strength, and removing the axial ambiguity of the system with respect to the parallel and antiparallel orientations of the external field relative to the

magnet's rest orientation). To first order, therefore, the high magnetic energy of the giant spearhead ($MB = 121$ $k_BT$) offers some immediate advantages in terms of magnetoreception performance, for a given value of $K$, compared to the minimum baseline case for candidate magnetoreceptive structures (taken here to be $MB \sim 6$ $k_BT$[52]). Further evidence of optimisation can be found by considering the case of fields applied antiparallel to the resting angle of the magnet[56] (Fig. 4b). In this configuration, when the ratio $MB/K$ approaches 1, the system operates at a critical point and $\Delta\psi$ changes rapidly as a function of field intensity for a fixed value of $K$. The thermal fluctuations cause a fluctuating transduced signal that can be used to measure the field intensity[23]. The sharpness of the field dependence of $\Delta\psi$ at the critical point is 2.5 times greater for $K = 121$ $k_BT$ compared to $K = 6$ $k_BT$ (Fig. 4b inset). Hence the giant spearhead with $MB = 121$ $k_BT$, attached to an elastic pivot with $K \sim 121$ $k_BT$, would show enhanced sensitivity to changing magnetic field intensity compared to a cluster of magnetic nanoparticles with $MB = 6$ $k_BT$ attached to a pivot with $K \sim 6$ $k_BT$. The large changes in $\psi$ that occur around the critical point can also be used to detect small angular deviations ($\eta$) from perfect antiparallel alignment with the field[29,56]. We calculate that the giant spearhead would provide a determination of the field direction to at least within $\eta \sim 1°$ (Supplemental Information), compared to $\eta \sim 10°$ for $K \sim 6$ $k_BT$. Hence, although the spearhead is primarily optimised for magnetointensity reception, it also delivers an accurate means to determine the field direction. We cannot rule out the presence of other magnetoreceptors in the animal that are optimised more specifically for just direction.

To address whether this optimisation in magnetoreception performance can be generalised to other giant magnetofossils, the field sensitivity of $\Delta\psi$ and the minimum detectable deviation angle at the critical point are plotted in Fig. 4c for a range of biologically plausible values of $K$. The calculated moments of 24 giant magnetofossils (Fig. S6) with a range of volumes and morphologies (including needles, giant bullets, kinked giant bullets, spearheads and spindles) are presented in Table S1 and as a histogram in Fig. 4c. The mean magnetic energy is 93 $k_BT$, with a median value of 36 $k_BT$. The maximum value is 523 $k_BT$ (for a spindle) and values cluster around 30 $k_BT$ for needles and giant bullets and around 100 $k_BT$ for spearheads. The two largest values both correspond to spindles. The distribution of calculated magnetic energies matches the range where the most rapid improvements in magnetoreception performance are seen (i.e., the greatest increase in $\Delta\psi$ and the greatest decrease in $\eta$). Further improvements in performance are possible by increasing magnetic remanence. For particles containing a single-vortex state, this could be achieved by increasing the overall size of the particle and the length of its remanence carrying core. However, the single-vortex structure in equidimensional magnetite is known to break down into a multi-domain state at volumes only slightly greater than those observed here[57], so there is a risk that larger spearheads could result in reduced stability and reproducibility of the remanence, to the detriment of magnetoreceptive performance. The clustering of magnetic energy values for spearheads around ~100 $k_BT$ might, therefore, be considered as striking the 'optimum' balance between improved performance and domain-state predictability (and, notably, is close to the mean value of all fossil types modelled here). Support for this principle of optimisation is found in the size distributions of giant magnetofossils[5]. For example, the remanent moment of needles increases by increasing their overall length whilst maintaining the very high aspect ratio needed to stabilise the uniformly magnetised state[58]. Once needles have reached an average diameter of 100 nm, they are observed to grow in length only[14], reaching values up 2 μm (corresponding to a magnetic energy of 92 $k_BT$ – again, similar to the mean of all fossil types). For giant bullets and spearheads, which both adopt single vortex states, the length and width of particles increase more in proportion to each other, up to a maximum length of <3–4 μm. The absence of giant magnetofossils larger than 3–4 μm is consistent with the desire to retain the single vortex state. Beyond the single particle scenario, the alternative strategy for increasing moment is by building ever larger clusters of uniformly magnetised magnetite nanoparticles[46]. However, this strategy may come at the expense of the domain state predictability offered by a single, giant magnetofossil, where

features such as vortex cores form in a natural and predictable way due to the interplay of shape anisotropy and magnetocrystalline anisotropy. In addition, the well-defined crystal terminations of giant magnetofossils (Fig. 1a) could provide a more controllable and reproducible means of anchoring the particle within the cell.

In summary, our observations and analysis support the hypothesis that single, giant magnetofossils could form viable magnetoreceptors, with moments large enough to operate close to their critical point whilst delivering optimal information regarding intensity and accurate directional information as a useful byproduct.

## Discussion

The ability to detect variations in both direction and intensity of Earth's magnetic field creates an evolutionary advantage for mobile animals, which would be able to develop an accurate navigational map sense[25,26]. Even in the case that the particle studied here evolved for the purposes of protection in some small worm, the possibility remains that it might have been exapted for a component of magnetointensity reception in a more migratory descendent. By optimising magnetoreception performance to detect ever smaller variations in intensity and direction, this navigational sense would become useful for small organisms that forage over relatively short distances (e.g., as seen in ants and bees[59]), i.e. not just large ones, such as fish, that navigate over thousands of kilometers. Whatever organism is responsible, it should be sufficiently abundant to explain the widespread occurrence of their fossilised remains in sediments (albeit at several orders of magnitude lower concentration than conventional magnetofossils), which would favour smaller culprits over larger ones. Such considerations may help narrow our search for the organism(s) responsible for creating giant magnetofossils, thereby revealing their true environmental and ecological significance. Although the presence of functional magnetoreception in extant mollusks, amphibians, fish, reptiles, birds, and mammals (among other groups) argues that magnetoreception dates back well before the Cambrian explosion (to at least the first bilaterians 550 Myr), if confirmed, our work would provide the first fossil evidence that navigational magnetoreception developed as a sense in eukaryotes at least 97 million years ago[4,6]. The enhanced preservation potential of giant magnetofossils compared to their conventional counterparts opens up the possibility that earlier evidence for the development of this sense may be present in the fossil record. The methods applied here demonstrate how evidence of biological optimisation in putative magnetofossils (or even in candidate receptor cells themselves) may be assessed through direct imaging of the 3D magnetic topology of natural iron oxide particles in the multi-micron size range. This capability will be of interest to geologists and palaeontologists searching for evidence of the earliest iron biomineralising organisms on Earth[60–62], and is a test case of what magnetic observations could be made to support evidence for or against a biogenic origin[63]. An exciting future application of this will be to assess the morphological and magnetic fingerprints of putative biogenic iron oxide particles, which are anticipated to be a key component in the search for evidence of past life in samples returned from Mars[64,65].

## Methods
### Sample collection and preparation
The marine sediment sample was obtained from International Ocean Discovery Program (IODP) North Atlantic Hole U1403B with the stratigraphic position at section 23X1W, 24–25 cm interval, within the Paleocene–Eocene Thermal Maximum period ( ~ 4374 m paleodepth)[5]. Magnetic mineral extracts were obtained following the method of[3]. First, ~2 cm³ wet samples were dispersed in ~100 ml of pure water in a glass beaker for ~15 min in an ultrasonic bath. A rare-earth magnet placed into a plastic bag was dipped into the sediment solution and stirred for ~3 min to gather magnetic extracts on the plastic bag surface. After placing this plastic bag into a larger one and removing the rare-earth magnet, magnetic extracts were collected in the large plastic bag. This process was repeated until magnetic extracts could barely be obtained from the sediment solution. The magnetic extracts attached to the plastic bag surface were washed with a

small amount of pure water. Second, the solution with magnetic extracts was transferred into a 5 ml centrifuge tube filled with pure water using a pipette. A rare-earth magnet was placed next to the centrifuge tube and positioned ~0.5 cm above the bottom of the tube. After ~2 h, non-magnetic materials deposited at the tube bottom were gently removed with a pipette, and magnetic materials trapped on the tube wall were collected. The separation for magnetic and non-magnetic materials was repeated three times to increase the concentration of magnetic minerals in the extracts. To prepare samples for SEM and X-ray ptychography imaging, magnetic materials with a small amount of pure water were first transferred into a small container. A TEM grid consisting of a 1 mm × 1 mm × 50 nm $Si_3N_4$ membrane in a 3 mm × 3 mm × 200 μm Si frame was placed on the solution surface. A rare-earth magnet was suspended over the grid ~1 cm for ~5 min. Several samples were prepared with the aim of finding a representative fossil close to the centre of the $Si_3N_4$ membrane, which facilitates data acquisition. The position of the particle on the membrane was identified using SEM.

### Synchrotron measurements

Soft X-ray magnetic tomography was performed at the I08-1 instrument of the Diamond Light Source. To probe the magnetic configuration of the giant magnetofossil, pre-edge phase dichroic imaging was performed with soft X-ray ptychography. The use of pre-edge photon energies facilitates X-ray transmission through the large sample (maximum projected thickness ~2 μm). Use of the phase XMCD signal, obtained by calculating the difference of phase images taken with left and right circular polarisation, provides much stronger magnetic contrast than the corresponding amplitude XMCD signal in the pre-edge region[32]. A Fresnel Zone Plate (diameter 333 μm, outermost zone width 70 nm) and Order Sorting Aperture were used to define a probe of size 1 μm on the sample. Coherent diffraction patterns were collected in the far field using an sCMOS detector (AXIS-SXRF-400EUV, Axis Photonique Inc., 11 μm pixel size, distance 72 mm, exposure time 40 ms, 2048 × 2048 pixels cropped to 1024 × 1024 and down sampled to 512 × 512) for a number of overlapping positions on the sample, by scanning the sample in a spiral path geometry. The full complex transmission function was then reconstructed using the GPU-accelerated python-based framework PtyPy[66,67]. Ptychographic projections were measured with circular right and left polarisation with XMCD calculated using the difference of the reconstructed phase projections measured with opposite circular polarisations. The resulting isolated magnetic contrast represents the integrated component of the magnetisation parallel to the X-ray direction. Measurements were initially performed as a function of energy across the Fe $L_3$ edge to maximise the XMCD contrast. Optimal imaging conditions corresponded to an X-ray energy of 706 eV, 4.5 eV below the observed peak of the absorption edge. To obtain the three-dimensional magnetic configuration, ptychographic projections were measured about two perpendicular axes of rotation. A first tomographic dataset was collected by measuring projections at angles −30° to +43° in 1° steps. To gain access to the third component of the magnetisation, a second tomographic measurement was performed after rotating the sample by 90° about the sample normal, and measuring a second set of tomographic projections, with angles −42° to +34° in 1–2° steps. A total of 73 and 71 projections were acquired about both axes of rotation, with an angular sampling corresponding to a maximum spatial resolution of approximately 50 nm. Note that due to the limited tilt angle range, a "missing wedge" effect likely means that the 3D reconstruction is artificially stretched along the beam direction (x), with an associated decrease in spatial resolution. However, the relative positions of the features identified are not affected.

### 3D reconstruction

To align the tomographic projections, nearest neighbour cross correlation was first implemented, followed by a sub-pixel vertical mass alignment[68]. The two tomographic datasets were aligned with respect to each other by cross correlating the 0° projections after a 90° rotation. The three-dimensional magnetisation vector field was then reconstructed from the aligned set of all tomographic projections using a GPU-implementation of the gradient-based iterative reconstruction algorithm[28]. The algorithm initializes with a guessed 3D object (initially zero-filled). A set of projections from this guessed 3D object are generated for each measured angle. An error metric is calculated by obtaining the sum squared difference between the guessed projection dataset and the experimentally measured projections. The error metric is minimised by updating the object at each iteration, guided by the gradient of the error metric with respect to all three components of the magnetization. The algorithm was run for 10 iterations. A mask was implemented to constrain the magnetisation vector field to the volume of the magnetic material. The resulting 3D magnetisation configuration was visualised using the open source software Paraview[69].

### Micromagnetic simulations

Finite-element micromagnetic simulations were performed using MERRILL[38] using values of the micromagnetic parameters for magnetite at room temperature (saturation magnetisation, $M_s$ = 4.80768 × 105 A/m; first cubic magnetocrystalline anisotropy constant, $K_1$ = −1.32658 ×104 J/m³; exchange constant, $A_{ex}$ = 1.33487 × 10⁻¹¹. A range of 3D models approximating the observed geometry of the particle (base radius 0.55 μm, tip radius 0.058 μm, length 2.25 μm) were simulated. More idealised models consisted of a cylindrical base and conical tip. More accurate models were created by tracing the two-dimensional outline of the particle from 2D projections, measuring the radius of the particle as a function of position along its length, and then building a 3D model from a series of 90 stacked circular or elliptical frustrums with height 25 nm each. For elliptical cross sections, the ratio of minor to major radii was set to 0.85, matching that estimated from the variation of 2D projected base radius as a function of projection tilt angle. Tetrahedral meshes were generated from the 3D models using a mesh resolution of 20 nm using Coreform Cubit software. Although this mesh resolution is a factor of two greater than the exchange length of magnetite, the choice was a necessary compromise to keep the generated mesh files and simulation times within practical limits given the available computing resources. One simulation at full 10 nm resolution was performed on a high-performance computing facility to confirm that the results were similar (Fig. S7). Simulations were performed with either [001], [111] or [113] aligned with the length of the particle. For [113] orientations with elliptical cross sections, simulations were performed with the [111] easy axis oriented in both the x-z plane (containing the long and short dimensions of the particle) and the y-z plane (containing the long and the intermediate dimensions of the particle). To generate local energy minimum states, energy minimisation was performed from random starting configurations. The global energy minimum state could be created by initialising using MERRILL's vortex initialisation option (core strength ± 2) followed by energy minimisation. The same state was obtained by simulating growth of the particle from either the tip down or the base up by adding one 25 nm thick frustrum at a time. Each new frustrum added was initialised with a random starting configuration and then the whole system was energy minimised. Growth models using 50 nm and 100 nm thick frustrums were also explored, yielding identical results. Hysteresis simulations were initialised with a saturated state along the applied field direction. Susceptibility simulations were initialised in the global energy minimum state and energy minimised in fields of ± 1 mT applied along x, y and z. The energy barrier for formation of a Bloch point was determined using the Nudged Elastic Band (NEB) method[70], as implemented in MERRILL, using 100 intermediate steps. The start and end states of the NEB calculation were the global energy minimum state with vortex core magnetised in positive and negative z directions, respectively. Simulation results were visualised using Paraview[69].

### Data availability

The raw data, reconstruction code, and visualisation files needed to reproduce this work have been made publicly available via Zenodo (https://doi.org/10.5281/zenodo.16838789).

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

## Acknowledgements

This study utilized samples from the International Ocean Discovery Program (IODP). Funding was received from the European Union's Research and Innovation Actions under grant agreements 101005611 and 101131765. R.J.H., P.Y.T., E.R. acknowledge funding from EXCITE (award number G106564) and EXCITE2 (award number G122171). L.C. and R.J.H. acknowledge funding from the Royal Society Newton Advanced Fellowship (grant NAF\R1\201096). L.C. acknowledges additional support from the National Natural Science Foundation of China (grant 42061130214). J.N. and C.D. acknowledge funding from the Max Planck Society Lise Meitner Excellence Program and funding from the European Research Council (ERC) under the ERC Starting Grant No. 3DNANOQUANT 101116043. J.N. acknowledges support from the International Max Planck Research School for Chemistry and Physics of Quantum Materials. L.M. acknowledges funding from Horizon Europe Programme via Marie Sklodowska-Curie fellowship (101067742) and funding from the Spanish Government (project no. MCINN-24-PID2023-150968OA-I00). This work was carried out with the support of Diamond Light Source, Instrument I-08 (proposals MG33254-1 and MG35037). We would like to acknowledge Dr. Simone Finizio (Paul Scherrer Institute, Villigen, Switzerland) for additional assistance with the synchrotron data collection and Rong Huang for assistance in locating giant magnetofossil crystals under scanning electron microscopic observation.

## Author contributions

Conception and design: S.V., R.J.H., C.D., L.C.; Provision of sample and sample preparation: L.C., P.X. Electron microscopy: P.X., P.Y.T., E.R.; Synchrotron data collection: J.N., S.V., R.J.H., R.A.; Synchrotron beamline support: B.K., B.D., L.C.C.H., M.K.; Synchrotron data processing: J.N., S.V., V.S.C.K., C.D., L.M.; 3D reconstruction: J.N., C.D.; 3D data visualisation: R.J.H., S.V., J.N.; Micromagnetic modelling: R.J.H., Z.P., L.M.; Interpretation and analysis: R.J.H., S.V., J.N., C.D., L.C., L.M.; Writing: R.J.H. with input from all authors.

## Funding

## Competing interests

The authors declare no competing interests.
