## [Transparent Peer Review file · Communications Earth & Environment]

Magnetic vector tomography reveals giant magnetofossils are optimised for magnetointensity reception

Corresponding Author: Professor Richard Harrison

Version 0:

Decision Letter:

Dear Professor Harrison,

Your manuscript titled "Giant magnetofossils are optimised for navigational magnetoreception" has now been seen by 3 reviewers, and we include their comments at the end of this message. They find your work of interest, but some important points are raised. We are interested in the possibility of publishing your study in Communications Earth & Environment, but would like to consider your responses to these concerns and assess a revised manuscript before we make a final decision on publication.

We therefore invite you to revise and resubmit your manuscript, along with a point-by-point response that takes into account the points raised. Please highlight all changes in the manuscript text file.

Please submit your point-by-point responses as a separate file, distinct from your cover letter where you can add responses to the Editors' comments that you do not want to be made available to the reviewers. Word files are preferred. We recommend that any figures, tables or graphs that are included in the response to reviewers are also included in the main article or Supplementary Information.

Please use the following link to submit your revised manuscript, point-by-point response to the referees' comments (which should be in a separate document to any cover letter), a tracked-changes version of the manuscript (as a PDF file) and the completed checklist:

Link Redacted

We hope to receive your revised paper within six weeks; please let us know if you aren't able to submit it within this time so that we can discuss how best to proceed. If we don't hear from you, and the revision process takes significantly longer, we may close your file. In this event, we will still be happy to reconsider your paper at a later date, as long as nothing similar has been accepted for publication at Communications Earth & Environment or published elsewhere in the meantime.

Please do not hesitate to contact us if you have any questions or would like to discuss these revisions further. We look forward to seeing the revised manuscript and thank you for the opportunity to review your work.

Best regards,

Joe Aslin

Deputy Editor,

Communications Earth & Environment

Consulting Editor,
Communications Sustainability

<https://www.nature.com/commsenv/>
Twitter: @CommsEarth

EDITORIAL POLICIES AND FORMATTING

Editorial Policy: [Policy requirements](https://www.nature.com/documents/nr-editorial-policy-checklist.pdf) (Download the link to your computer as a PDF.)

- Behavioural and social science
- Ecological, evolutionary & environmental sciences
- Life sciences

<https://www.nature.com/documents/nr-reporting-summary.zip>

Furthermore, please align your manuscript with our format requirements, which are summarized on the following checklist: [Communications Earth & Environment formatting checklist](https://www.nature.com/documents/commsj-phys-style-formatting-checklist-article.pdf)

and also in our style and formatting guide [Communications Earth & Environment formatting guide](https://www.nature.com/documents/commsj-phys-style-formatting-guide-accept.pdf).

*** DATA: Communications Earth & Environment endorses the principles of the Enabling FAIR data project (<http://www.copdess.org/enabling-fair-data-project/>). We ask authors to make the data that support their conclusions available in permanent, publically accessible data repositories. (Please contact the editor if you are unable to make your data available).

All Communications Earth & Environment manuscripts must include a section titled "Data Availability" at the end of the Methods section or main text (if no Methods). More information on this policy, is available at <http://www.nature.com/authors/policies/data/data-availability-statements-data-citations.pdf>.

If a community resource is unavailable, data can be submitted to generalist repositories such as [figshare](https://figshare.com/) or [Dryad Digital Repository](http://datadryad.org/). Please provide a unique identifier for the data (for example a DOI or a permanent URL) in the data availability statement, if possible. If the repository does not provide identifiers, we encourage authors to supply the search terms that will return the data. For data that have been obtained from publically available sources, please provide a URL and the specific data product name in the data availability statement. Data with a DOI should be further cited in the methods reference section.

REVIEWER COMMENTS:

Reviewer #1 (Remarks to the Author):

Review of "Giant Magnetofossils are optimized for navigational magnetoreception", by Richard J. Harrison, et. al. Review by Joseph L. Kirschvink.

Summary:

The authors provide a tour de force micromagnetic analysis of the internal magnetic structure of enigmatic giant magnetofossils that were first described by Schumann et al.[1] from sediments at the Paleocene/Eocene thermal maximum in New Jersey. They conclude that these magnetic structures could serve a role in magnetoreception, in contrast to the hypothesis that it might have been a protective structure in some microorganisms that lived in these peculiar reducing environments.

Overall comments:

I was initially confused about this manuscript because it is very clear that magnetite crystals that naturally have only 2% of the saturation magnetization are certainly not optimized for magnetic compass reception. That is a gross waste of iron for any organism, and iron is one of the limiting nutrients in many marine ecosystems. In the magnetotactic bacteria it is very clear that they go to great lengths to maximize the contribution to the total cellular magnetic moment from every atom of iron that they can grub from the environment (e.g., [2]). There's a huge literature on this dating back nearly 50 years[3,4]. I think the authors mean that these large magnetofossils might be more suitable for the other known type of magnetic sensing - magnetointensity reception[5]. I would suggest modifying the title of the paper to state that perhaps giant magnetofossils might play a role in the detection of magnetic intensity fluctuations in the environment, but certainly not as simple compass needles. The idea of two separate forms of magnetoreception goes back 40 years[5].

In this regard, I am surprised that the authors do not mention the several instances reported by Schumann et. al[1] of clusters of these spear-like magnetofossils, apparently in an intact orientation from the original (but as yet unidentified) organism. These are clusters of up to 100 of these ~3 μm crystals, all with the spear-tip points radiating outward from a central core. These are not random aggregations of particles that have clumped together - if they were random clumps then you would expect a disordered array with the crystals pointing randomly, like a stack of needles. These structures have all of the tips pointing outwards, as demonstrated elegantly in the Schumann et al.[1] paper by using a 3 dimensional TEM tilt series (Supplemental video is on line at: https://www.pnas.org/doi/suppl/10.1073/pnas.0803634105/suppl_file/sm1.mov). Packed crystals like that could either be pointed inwards or outwards, so the chance probability of all 100 tips pointing outwards would be 2^{-100} , or $p < 1\text{E-}30$. However, the presence of many isolated giant magnetofossils that are clearly disaggregated from these clusters implies that some physical force in the environment must be able to overcome the magnetostatic binding energy between crystals in those clusters, even when in vivo. This was a puzzle for us when we first found them, as the initial TEM magnetic observation suggested they were almost single magnetic domains (but we did note that the magnetic extraction process would probably leave the particles with the transient IRM[1]). New data from this present manuscript reduces that problem substantially, as they show that the natural magnetic moments are only about 2% of the saturation value. However, I calculate that their new estimate of the magnetic moment of these particles (10^{-14} Am^2) still implies a particle-particle binding energy of ~ 700 kT, assuming an interparticle separation of 1.5 μm . Hence some process must still act to disrupt the residual clumping forces between crystals in the original clusters. I think this is most easily done if they were an abrasive armament on the skin of some worm-like creature in the mud, where shear forces could disrupt them. Iron-rich dermal spicules are well known in biology and can be found disaggregated in sediments[6-8]. It is thus not implausible that these magnetofossils are some sort of spicule like the dermal granules that Lowentam and Rossman (1975) reported in the skin of Holothurians[8]. Indeed, magnetite is biomineralized in the major lateral teeth of the Polyplacophoran mollusks (the chitons)[9], and used as a hardening agent to scrape endolithic algae from rocks. In turn, this leads to another puzzle as to how these particles could be so common in these particular dysoxic sediments. The worm (or whatever) must be fairly common to produce a high density of these objects in the sediment.

None of the above reasoning, however, argues against the conclusions in the present manuscript that novel particles related to these giant magnetofossils might actually be evolved in magnetointensity detection. That is an elegant concept, even if it is unlikely in the organism that left these particular fossils. Small worms in this environment would have been unlikely to be migratory, and so probably would not need to have a magnetointensity receptor in an environment where spatial magnetic fields don't change much and little oxygen is available. My reasoning is as follows: In the initial suggestion that biogenic magnetite might be involved in the intensity component of magnetoreception, Mike Walker and I [5] developed the "thermally driven variance model of magnetointensity reception". This simply demonstrated that neural signals from magnetosomes attached to mechanoreceptors could convey both directional information and intensity information to an animal's brain. The trick there was to realize that strong fields damp Brownian motion by increasing the magnetic to thermal energy ratio ($\mu\text{B}/k\text{T}$), pinning the moment to the background field direction, whereas thermal agitation and large angular motions dominate in weak fields. Hence, the variability in the action potentials firing rate from these cells is a monitor of total field sensitivity, whereas the mean firing rate would yield compass information. By analyzing the spatial derivatives of the Langevin function, we[5] concluded that the optimum sensitivity to magnetic intensity fluctuations would be at relatively small SD crystals, with $\mu\text{B}/k\text{T}$ ratios of about 2. Winklhofer and I[10] updated this analysis to consider in the context of mechanical torques that might be needed to produce such effects, but the basic principle is the same. Thus, the suggestion in this manuscript that the extraordinarily high magnetic susceptibility estimated for these giant magnetofossils could offer a different magnetosome-based path to magnetointensity reception is a novel idea. I would think that once these things evolved in some small worm, it might have been exapted for a component of magnetointensity reception in a more migratory descendent.

In summary, I recommend publication of this important study following revisions along the above thread, and in response to the comments below.

More detailed comments on text:

Line 78: It is *not* optimized for detecting magnetic field direction. Iron is one of the principal limiting nutrients in biology, and building a compass needle that is only 2% efficient is a waste. I strongly recommend removing these misleading statements throughout the ms.

Lines 134-135: Could the Bloch Point singularity described here be caused by a spiral screw dislocation in the magnetite crystal lattice? A spiral screw dislocation along the particle length would help the biomineralization process, but it has never been documented in HRTEM analyses of bacterial magnetite. Does your data on the natural crystals rule this out?

Lines 193-196, and at the end of the supplemental information. I am not surprised that the vortex geometry of these crystals would produce such a high intrinsic magnetic susceptibility. However, the analysis would be much more powerful if you calculated both the real and imaginary components of the complex magnetic susceptibility for the giant magnetofossils and other particles and add them to Table S2. That would help us compare the intrinsic properties to past studies of superparamagnetic magnetite [11], for example. It would also contribute greatly if you could calculate the imaginary component (X'' , that controls energy dissipation) out to a few GHz. These particles might have applications in the ferrite industry, and such calculations might lead to inorganic synthesis and potential industrial applications. I'm not familiar enough with the MERRILL software to know if that is feasible.

Line 205-206, Again, this is an inefficient use of metabolic iron.

Lines 220-221: "The eukaryotic cells are interesting in bridging the gap between magnetotactic bacteria ... and origin in Eukaryotes." Hey, this is precisely the proposal that Vali and I [12] made in 1991. (A PDF is on my www site if you don't have the book.)

Lines 229 to 238: I have a few comments here. First, direct extracts followed by TEM analysis of the olfactory tissues of trout and salmon (genus *Oncorhynchus*) actually revealed magnetosome chains which were very similar to those from the magnetotactic bacteria, including {111} crystal twins (see Mann et al., 1988 [13]). The MFM technique of Diebel et al. [14] (2000, ref #45) identified clusters, but could only map out the minimum size of the magnetite-containing cells, as the attraction of the probe dropped off as $1/r^4$. There was no 3-D information in that analysis. Hence, the best data for the moment of these structures is from the second study you cite, Eder et al. (2012, ref. #46), where the whole cells were rotating. There is no evidence for two size categories of the magnetite crystals, and the TEM analysis of Mann et al. [13] of the olfactory extracts from these tissues most certainly would have identified the giant magnetosomes if they had been present in the olfactory epithelium. The 1997 paper by Walker et al. [15] demonstrated that magnetite-containing cells in that tissue were connected to magnetically-responsive nerve fibers in the supraorbital trunk, using a lipophilic dye, so it is clear that they are magnetoreceptor cells. Eder et al. [16] even showed that rotating the magnetic field on stabilized cells caused Ca^{++} ion channels to open. Note that the Edelman et al. (2015) [17] attempt to replicate the Eder et al. [16] rotating cell paper should just be ignored, as they failed to follow well-known protocols for removing contamination in the reagents and chemicals used for disaggregating the trout olfactory epithelium. They apparently had not seen the 1995 Nature paper by Kobayashi et al. [18] showing that such contaminants were ubiquitous in common 'reagent-grade' chemicals and solutions. It is therefore no surprise that Edelman et al. only found contaminants. As you note, Bellinger et al. (ref. #39) replicated the clean-lab extraction of these magnetic cells from trout and made a good start on the genomics, indicating that the cells really do exist.

My conclusion is that if giant magnetosomes like the one Harrison et al. have modeled are part of the magnetointensity receptor complex in higher animals, they are not co-located with the compass receptor cells. Finding them is a worthwhile goal.

Line 258 – again, Ref. #45 is an under-estimate.

Lines 269-270. The nervous system is very good at integrating small signals from a large number of receptor cells to extract a small signal from the background and doing pattern recognition; this is how a parent can pick up the tiny, faint sounds of their child at day care against the background of all of the other children (at least I could). Thus, if one large spearhead could be accurate to 1° with a mass 300x that of a 6 kT magnetosome chain of accuracy 10° , I would say that an array of 300 of the 6 kT chains would give the brain an accuracy of $10/\sqrt{300}$, or 0.56° . The analysis presented here does not make sense in terms of how the nervous system works.

Lines 272-307, and Fig. 4. This is definitely a variant of the 'thermally-driven variance model' of magnetoreception⁵, and should be cited as such. The difference is that you include a rigidity of the attachment fibers that changes the tuning of the system. However, I am still not convinced that there would be an advantage of a crystal of magnetite of only 2% saturation, compared with an assemblage of smaller crystals which could achieve the same magnetic moment with correspondingly less iron required (see above). You should also go into the literature in neurobiology concerning the two major categories of mechanical neuroreceptor cells – phasic and tonic (google it!). The well-known difference lends itself nicely to the understanding of the two forms of magnetoreception – phasic cells could easily determine a compass direction, whereas the tonic cells could monitor the intensity.

Lines 347-348. Ah, the presence of functional magnetoreception in extant mollusks, amphibians, fish, reptiles, birds, and mammals (among other groups) argues that magnetoreception dates back well before the Cambrian explosion (to at least the first bilaterians 550 Myr).

I look forward to seeing a revision!

J.L. Kirschvink

References cited in the review (mostly background information for the Authors):

- 1 Schumann, D. et al. Gigantism in unique biogenic magnetite at the Paleocene-Eocene Thermal Maximum. Proceedings of the National Academy of Sciences of the United States of America 105, 17648-17653 (2008). <https://doi.org/10.1073/pnas.0803634105>
- 2 Kirschvink, J. L. Paleomagnetic evidence for fossil biogenic magnetite in western Crete. Earth & Planetary Science Letters 59, 388-392 (1982).
- 3 Kirschvink, J. L. & Gould, J. L. Biogenic magnetite as a basis for magnetic field sensitivity in animals. Bio Systems 13, 181-201 (1981).

- 4 Kirschvink, J. L. & Lowenstam, H. A. Mineralization and magnetization of chiton teeth: Paleomagnetic, sedimentologic, and biologic implications of organic magnetite. *Earth & Planetary Science Letters* 44, 193-204 (1979).
- 5 Kirschvink, J. L. & Walker, M. M. in *Magnetite Biomineralization and Magnetoreception in Organisms: A New Biomagnetism Vol. 5 Topics in Geobiology* (eds J.L. Kirschvink, Jones D.S., & B. McFadden) 243-254 (Plenum Press, 1985).
- 6 Lowenstam, H. A. Minerals made by organisms. *Science* 211, 1126-1131 (1981).
- 7 Lowenstam, H. A. & Weiner, S. *On Biomineralization*. (Oxford University Press, 1989).
- 8 Lowenstam, H. A. & Rossman, G. R. AMORPHOUS, HYDROUS, FERRIC PHOSPHATIC DERMAL GRANULES IN MOLPADIA (HOLOTHUROIDEA) - PHYSICAL AND CHEMICAL CHARACTERIZATION AND ECOLOGIC IMPLICATIONS OF BIOINORGANIC FRACTION. *Chem. Geol.* 15, 15-51 (1975).
- 9 Lowenstam, H. A. Magnetite in Denticle Capping in Recent Chitons (Polyplacophora). *Geological Society of America Bulletin* 73, 435-& (1962). [https://doi.org/Doi.10.1130/0016-7606\(1962\)73\[435:Midcir\]2.0.Co;2](https://doi.org/Doi.10.1130/0016-7606(1962)73[435:Midcir]2.0.Co;2)
- 10 Winklhofer, M. & Kirschvink, J. L. A quantitative assessment of torque-transducer models for magnetoreception. *J R Soc Interface* 7 Suppl 2, S273-289 (2010). <https://doi.org/10.1098/rsif.2009.0435.focus>
- 11 Malaescu, I. & Marin, C. N. Study of magnetic fluids by means of magnetic spectroscopy. *Physica B: Condensed Matter* 365, 134-140 (2005). <https://doi.org/https://doi.org/10.1016/j.physb.2005.05.006>
- 12 Vali, H. & Kirschvink, J. L. in *Iron Biomineralization* (eds R.P. Frankel & R.P. Blakemore) 97-115 (Plenum Press, 1991).
- 13 Mann, S., Sparks, N. H., Walker, M. M. & Kirschvink, J. L. Ultrastructure, morphology and organization of biogenic magnetite from sockeye salmon, *Oncorhynchus nerka*: implications for magnetoreception. *J Exp Biol* 140, 35-49 (1988).
- 14 Diebel, C. E., Proksch, R., Green, C. R., Neilson, P. & Walker, M. M. Magnetite defines a vertebrate magnetoreceptor. *Nature* 406, 299-302 (2000). <https://doi.org/10.1038/35018561>
- 15 Walker, M. M. et al. Structure and function of the vertebrate magnetic sense. *Nature* 390, 371-376 (1997).
- 16 Eder, S. H. et al. Magnetic characterization of isolated candidate vertebrate magnetoreceptor cells. *Proc Natl Acad Sci U S A* 109, 12022-12027 (2012). <https://doi.org/10.1073/pnas.1205653109>
- 17 Edelman, N. B. et al. No evidence for intracellular magnetite in putative vertebrate magnetoreceptors identified by magnetic screening. *Proc Natl Acad Sci U S A* 112, 262-267 (2015). <https://doi.org/10.1073/pnas.1407915112>
- 18 Kobayashi, A. K., Kirschvink, J. L. & Nesson, M. H. Ferromagnets and EMFs. *Nature* 374, 123-123 (1995).

Reviewer #2 (Remarks to the Author):

Please see my review, attached.

Reviewer #3 (Remarks to the Author):

The authors leverage magnetic vector field tomography to visualize the magnetization in a giant spearhead magnetofossil and use an analytical equation to assess its magneto receptive response. The introduction is great for a general audience; the results may be of interest to the readership of *Communications Earth & Environment* following major revision.

Clearly state the energy used for pre-edge XMCD measurements (and shift from resonance) at the beginning and also include in caption. Please add details for "XMCD projections acquired about two rotation axes", e.g., y and z axis with x-rays incident along x axis.

The authors show the crystallographic orientation and state magnetocrystalline values. How is the crystal orientation determined? I couldn't find any data on this. How is the magnetocrystalline considered in both experiment and simulation?

Taking into account the authors' statement "These results suggest that the Bloch point and tip domain are likely caused by exposure of the sample to a magnetic field during magnetic extraction from the sediment (see Methods), rather than being primary features of the fossil's natural, lowest energy magnetic state.", how can they be sure that not the entire magnetization configuration has been altered? If this was a likely scenario, would it somewhat affect their conclusions/key outcomes?

What magnetization value M was used as pertaining to "In the context of magnetotaxis, the alignment efficiency can be quantified by the ratio of magnetic to thermal energy (MB/kBT , where M is the magnetic moment in Am^2 , B is the Earth's field in Tesla, kB is Boltzmann's constant and T is temperature)." Was this the saturation magnetization of the remanent magnetization or the net moment of the elongated vortex core or the adjusted magnetization at the applied magnetic field as determined from hysteresis loops? It can surely not be the former.

Given the S-shape of the hysteresis compared with the square shape of nanocubes, it is not surprising to obtain a higher susceptibility for the former. However, we would that be beneficial? A rigid moment should be superior in terms of magnetoreception.

The authors compare their magnetic state with literature: "The dominant feature is a single vortex with curved/kinked core trajectory (Fig. 2a), rather than the multi-domain state previously predicted for slightly larger spearheads with a stalk^{2,14} or the uniform state previously claimed on the basis of 2D electron holography observations¹." While this is appropriate I recommend rephrasing. Clearly the samples must have been different. There is no way electron holography would be misinterpreted that strongly. To this extent, the results should be presented as corroboration of highly sensitivity of shape (or similar, please elaborate).

The change in sign of the normal magnetization in the tip domain is puzzling. What is the physical interpretation? I would have expected a conical spin texture with the direction along the vortex core and/or external field.

How does the reconstruction converge? What metric is used? Ten iterations seem very few.

I am a bit confused by the authors' statement "Although this mesh resolution is a factor of two greater than the exchange length of magnetite, the choice was necessary to keep the generated mesh files and simulation times within practical limits, and is typically considered an acceptable compromise." The mesh resolution is excessively large and would be unacceptable to the magnetism community where the mesh size should be half the exchange length, not twice its value.

Please elaborate on the "The methods developed here". To my best knowledge, the presented methods were simply applied not developed.

I suggest rephrasing "oriented predominantly normal to z, i.e., parallel to the basal plane of the particle and going into and out of the plane of the diagram in each half." As "predominantly lay in the plane perpendicular to the symmetry axis of the spearhead" or something similar.

Communications Earth & Environment is committed to improving transparency in authorship. As part of our efforts in this direction, we are now requesting that all authors identified as 'corresponding author' create and link their Open Researcher and Contributor Identifier (ORCID) with their account on the Manuscript Tracking System prior to acceptance. ORCID helps the scientific community achieve unambiguous attribution of all scholarly contributions. You can create and link your ORCID from the home page of the Manuscript Tracking System by clicking on 'Modify my Springer Nature account' and following the instructions in the link below. Please also inform all co-authors that they can add their ORCIDs to their accounts and that they must do so prior to acceptance.

Version 1:

Decision Letter:

Dear Professor Harrison,

Your manuscript titled "Magnetic vector tomography reveals giant magnetofossils are optimised for magnetointensity reception" has now been seen by our reviewers, whose comments appear below. In light of their advice we are delighted to say that we are happy, in principle, to publish a suitably revised version in Communications Earth & Environment.

We therefore invite you to revise your paper one last time to address the remaining concerns of our reviewers. At the same time we ask that you edit your manuscript to comply with our format requirements and to maximise the accessibility and therefore the impact of your work.

EDITORIAL REQUESTS:

****Please take care to match our formatting and policy requirements. We will check revised manuscript and return manuscripts that do not comply. Such requests will lead to delays. ****

SUBMISSION INFORMATION:

In order to accept your paper, we require the files listed at the end of the Editorial Requests Table; the list of required files is also available at <https://www.nature.com/documents/commsj-file-checklist.pdf> .

OPEN ACCESS:

Communications Earth & Environment is a fully open access journal. Articles are made freely accessible on publication. For further information about article processing charges, open access funding, and advice and support from Nature Portfolio, please visit <https://www.nature.com/commsenv/open-access>

Link Redacted

Best regards,

Joe Aslin

Deputy Editor,
Communications Earth & Environment

Consulting Editor,
Communications Sustainability

<https://www.nature.com/commsenv/>
Twitter: @CommsEarth

REVIEWERS' COMMENTS:

Reviewer #1 (Remarks to the Author):

I am happy with the responses and revisions.
One minor fix - as far as I know, the chiton magnetite is still single crystals, not polycrystalline.

Reviewer #2 (Remarks to the Author):

I remain concerned about the reproducibility of the work on other magnetofossils and by other researchers. However, as applied here, the findings from this method have direct implications for the magnetofossil community. The manuscript looks to be in good shape after this revision so I do not have additional major comments.

Reviewer #3 (Remarks to the Author):

The authors' response and revision are satisfactory to warrant publication in CEE.

I would suggest adding a quantitative comparison for the micromagnetic simulations using 20 and 10 nm mesh size to the SI. This should include energy contributions and line profiles.

** Visit Nature Portfolio's author and referees' website at <http://www.nature.com/authors> for information about policies, services and author benefits**

Harrison and coauthors combine a three-dimensional magnetic vector tomography study with micromagnetic and torque-induced simulations/calculations on a giant spearhead-shaped magnetofossil. They use these data to support the case that giant magnetofossils are biogenic and engineered for magnetoreception. More specifically, they argue that a giant magnetofossil is enough for an organism to navigate using Earth's magnetic field, and perhaps evolutionarily favorable.

This manuscript has a lot to it, and I am impressed the authors were able to parse all this information into a compact manuscript and story. The authors show it is possible to directly measure and visualize the internal magnetic structure of giant magnetofossils. This is novel and impactful. This is really neat. Their method has potential for groups studying giant magnetofossils, Early Earth, and life on other planets. Unfortunately, as written, I think this message gets muddled and overshadowed by some of the interpretations being made about giant magnetofossils as a whole (especially since this manuscript focuses on a single spearhead and does not look in detail at other morphologies). With some restructuring and focusing of the manuscript toward the authors' method (which I think is the most impactful part of the manuscript), and the subsequent importance of it with proposed interpretations separated, I think it may be suitable for publication in *Communications Earth and Environment*.

Comments and requested revisions

1. I think this method and investigation of giant magnetofossils is significant and impactful. However, I was put off by the overall tone and claims from the manuscript. I don't think you meant to do this, but there seems to be a lot of overarching, sweeping statements about all giant magnetofossils in general (see line-by-line comments below). I encourage you to focus on the cool thing about your work, the method and findings, and less-so about making these huge statements until making the end of the manuscript when you make interpretations. Also, please make it more clear that many of these statements and interpretations are on giant spearheads, based on this study, before extending them to other giant magnetofossil morphologies. I recommend really highlighting what makes your paper so distinct from other papers that have discussed the size and domain state of spearheads before. It would also be helpful if you included headers and split the manuscript into more distinct themes (e.g., "Modeling the magnetic behavior" vs "Explanation for single-particle magnetoreception"...)
2. Again, this method combination is interesting, unique, and envelope-pushing. It is exciting to learn and read about. I am not an expert in these measurements and techniques, but I am a little concerned about the reproducibility of these datasets.

Only one spearhead was directly measured and that does not represent natural variability between spearheads and other giant magnetofossil morphologies. For example, you do not fully consider that many spearheads are found with stalks attached to them, which is sure to effect the magnetic characteristics of the spearhead too (see your supplemental figure 6 with simulations). To this point, there is actually a wide range of proposed magnetic domain states of spearheads due to the varying sizes (Wagner et al., 2024; Xue, Chang, Pei, et al., 2022; Xue & Chang, 2024). Please bring in more discussion about the spearhead variation and caveats associated with this. For example, what role do you think the stalks play/how do you think they would influence the magnetic domain state of the spearheads? In future work I would implore you to strengthen this study with replications on other spearhead morphologies (e.g., more/less elongated and with/without stalks), other giant magnetofossils morphologies, and from different time periods and sediments to account for natural variability.

3. The spearhead studied here was magnetically extracted from the sediment such that it is not in its natural magnetic state of “life” position. As noted in the manuscript, it is also surrounded by other detrital particles which were magnetically extracted and have magnetization too. You also note that the spearhead has a high susceptibility. All these things make me concerned about whether you are capturing the “true” magnetic state of the spearhead. Please address these concerns within the manuscript with caveats or supportive reasoning. As another aside, how do you think having such a high susceptibility (especially compared to MTB) is beneficial for magnetotaxis? How does this compare to organisms that we know make magnetic particles, but not for magnetoreception (e.g., dissimilatory reducing bacteria, sclerite from gastropods)? Comparing to these organisms could help bolster your arguments.
4. Another note on reproducibility: this method seems complicated. Will other researchers, like the ones you discuss in the manuscript (paleontologists, etc.), really be able to reproduce and easily implement this method to directly measure giant magnetofossils? What will it take to make it happen?
5. My first thought when you started to describe the Bloch point was that it was going to be a descriptive feature of magnetofossils. I worried about bias from magnetic extraction, but then you addressed that. I’m a bit confused why there is so much emphasis on the Bloch point in the manuscript then, especially since you state that it’s unlikely a natural signature? I think better/more clear organization of the manuscript could help clear this up. For example, if you are simply trying to describe the magnetic characteristics you identified, consider putting all of this into a section

called “Identifying magnetic characteristics from direct measurements and simulations” or something.

6. This is picky, but the term “magnetosome” refers to the entire organelle that encompasses the magnetic particles. In general, please only use magnetosome when referring to the entire organelle. This will satisfy any MTB microbiologists reading your paper.

Lines 24-27: This comes off very strong, like we have a complete consensus that giant magnetofossil are biogenic. I agree that they are likely biogenic, but I encourage you to make this more conservative. Until we find the organism responsible for making them, we cannot say for certain.

Lines 27-30: This is also written very strongly, like there is a consensus between everyone that the main function of giant magnetofossils is for protection, but that is not true (Wagner, Egli, et al., 2021; Wagner, Lascau, et al., 2021; Xue, Chang, Dickens, et al., 2022; Xue & Chang, 2024). Other groups have suggested this “alternative” hypothesis already, that they are used for magnetoreception (see the same references as examples). Please change the wording in these sentences to better reflect the variety of interpretations already put out there, for e.g., “The most common interpretation is that ...” “We provide more evidence in favor of the hypothesis that that giant magnetofossil producing organisms exploited...”

Lines 31-34: Do “all” giant magnetofossils show the same behaviors/magnetic traits? I think you mean that to say this for spearhead-shaped fossils only.

Line 36: Why just eukaryotes? There are prokaryotes large enough to “house” them. Again, I have some issues with these strong statements since we do not know what organism made them. Please add another brief sentence here to make the connection between why these magnetic traits and navigational magnetoreception earlier in the fossil record.

Line 38: iron oxide

Lines 44-47: Reword or break into two separate sentences

Lines 48-51: Please add the corresponding references for each of these evidences in your sentence. Also, what about iron isotopes, growth structure, and magnetic signatures (Amor et al., 2016, 2018; Mathon et al., 2024; Wagner, Egli, et al., 2021; Xue, Chang, Pei, et al., 2022)?

Lines 55-60: Do you mean to say that this is just with respect to spearheads? Otherwise, I have similar comments to those above about the abstract. Please reword this to better reflect the variety of interpretations out there for all giant magnetofossils. There is no

general consensus and a lot of people already agree that they were used for magnetoreception. Also, most people say that the organism(s) responsible are likely eukaryotes because of the size of giant magnetofossils, but it's still not impossible that they are/were prokaryotes.

Lines 60-65: Break this up into a couple of sentences

Lines 70-75: Micromagnetic simulations were performed on giant needles imaged using TEM and their corresponding FORC signatures were modeled from this too (Wagner, Egli, et al., 2021). Please clarify that magnetic studies haven't been done directly on giant spearheads, which are the main focus of your study.

Line 75-76: This is the really cool, important, and new thing that your manuscript brings to the table: a way to directly look at the internal magnetic structure of a giant magnetofossil. I recommend emphasizing and focusing on this as your main takeaway.

Lines 80-85: Please include a brief sentence of how you obtained the particle. Did you magnetically extract it from the sediment? Why this one? There's a disconnect because the methods section is separated in this format.

Lines 187-227: In addition to comparing to magnetite biomineralized by MTB, which we know perform magnetotaxis, I recommend adding some comparisons to the magnetic properties of iron particles made by other organisms for purposes other than magnetotaxis and magnetoreception (e.g., protection, which you are arguing against in this paper), as negative supporting arguments (e.g., Fe-reducing bacteria that facilitate magnetite precipitation, the scaly-foot snail, etc.)

Line 190: consider a brief sentence relating to the remnant moment of MTB magnetite chains

Line 191: ...coercivity $B_c \sim 2\text{mT}$ (40 times greater than the Earth's field) and coercivity..."

Line 206: replace magnetosomes with something else... "magnetic particles made by MTB"

Lines 215-219: This (their remanent moment) seems the most compelling evidence to me that they could have been used by another, larger organism like the ones you compare them to (MMPs and eukaryotes). I recommend comparing this to abiotic magnetite crystals to be safe/strengthen your biogenic argument

Line 219: I think you mean magnetic particles as "magnetosome" refers to the entire organelle

Line 226 and 239: magnetic particle, not magnetosome

Line 252: this part about weaker field strengths seems to be something you can test

Lines 291-295: How do these magnetic energies relate to magnetite chains made by MTB and the protists and eukaryotes that we know use magnetoreception in the modern?

Line 296-298: I agree that whoever made them produced them through controlled biomineralization to prevent them being too big or small, but be careful because there are some reported larger than 3 microns (Xue, Chang, Pei, et al., 2022)

Lines 299: How are you predicting the domain state stability for all giant magnetofossils? Is this based on your magnetic energy calculations or is this just with respect to giant spearheads?

Line 303: What about the role of the stalks? Could those be helpful in “anchoring”?

Line 303 and 305: please use magnetofossils or crystals, not magnetosomes

Lines 341-345: Please be careful with this statement. Previous studies have shown that there are actually very few giant magnetofossils and it can be difficult to find them at all (Kadam et al., 2024; Wagner et al., 2024; Wagner, Lascu, et al., 2021)

Line 348: References: Wagner et al., 2024; Xue & Chang, 2024

Line 517: Why/how did you choose this one spearhead? (besides it being in the center of membrane)

Supplemental Figure S6: I recommend blending in more of the spindle variety simulations into your main text discussion about optimization of your measured spindle with respect to different lengths/widths and stalks

References cited in review

Amor, M., Busigny, V., Louvat, P., Gélabert, A., Cartigny, P., Durand-dubief, M., et al. (2016). Mass-dependent and -independent signature of Fe isotopes in magnetotactic bacteria. *Science*, 352(6286), 705–707. <https://doi.org/10.1126/science.aad7632>

Amor, M., Busigny, V., Louvat, P., Tharaud, M., Gélabert, A., Cartigny, P., et al. (2018). Iron uptake and magnetite biomineralization in the magnetotactic bacterium *Magnetospirillum magneticum* strain AMB-1: An iron isotope study. *Geochimica et Cosmochimica Acta*, 232, 225–243. <https://doi.org/10.1016/j.gca.2018.04.020>

Kadam, N., Badesab, F., Lascu, I., Wagner, C. L., Gaikwad, V., Saha, A., et al. (2024). Discovery of late Quaternary giant magnetofossils in the Bay of Bengal. *Communications Earth and Environment*, 5(1), 1–12. <https://doi.org/10.1038/s43247-024-01259-0>

Mathon, F. P., Amor, M., Guyot, F., Menguy, N., Lefevre, C. T., & Busigny, V. (2024).

Establishing the content in trace and minor elements of magnetite as a biosignature of magnetotactic bacteria. *Geochimica et Cosmochimica Acta*, (September).
<https://doi.org/10.1016/j.gca.2024.09.020>

- Wagner, C. L., Lasca, I., Lippert, P. C., Egli, R., Livi, K. J. T., & Sears, H. B. (2021). Diversification of iron-biomineralizing organisms during the Paleocene-Eocene Thermal Maximum: evidence from quantitative unmixing of magnetic signatures of conventional and giant magnetofossils. *Paleoceanography and Paleoclimatology*, 36(5), 1-25 e2021PA004225. <https://doi.org/10.1029/2021PA004225>
- Wagner, C. L., Egli, R., Lasca, I., Lippert, P. C., Livi, K. J. T., & Sears, H. B. (2021). In situ magnetic identification of giant, needle-shaped magnetofossils in Paleocene-Eocene Thermal Maximum sediments. *Proceedings of the National Academy of Sciences of the United States of America*, 118(6), 1-44 e2018169118. <https://doi.org/10.1073/pnas.2018169118>
- Wagner, C. L., Lasca, I., Self-trail, J. M., Gooding, T., Livi, J. T., Greger, G., et al. (2024). Discovery of giant and conventional magnetofossils bookending Cretaceous Oceanic Anoxic Event 2. *Communications Earth & Environment*, 5(1), 1–11. <https://doi.org/10.1038/s43247-024-01540-2>
- Xue, P., & Chang, L. (2024). Spatiotemporal distribution of giant magnetofossils holds clues to their biological origin, 52(6), 453–457. <https://doi.org/10.1130/G51809.1/6281751/g51809.pdf>
- Xue, P., Chang, L., Dickens, G. R., & Thomas, E. (2022). A Depth-Transect of Ocean Deoxygenation During the Paleocene-Eocene Thermal Maximum: Magnetofossils in Sediment Cores From the Southeast Atlantic. *Journal of Geophysical Research: Solid Earth*, 127(8). <https://doi.org/10.1029/2022JB024714>
- Xue, P., Chang, L., Pei, Z., & Harrison, R. J. (2022). Discovery of giant magnetofossils within and outside of the Palaeocene-Eocene Thermal Maximum in the North Atlantic. *Earth and Planetary Science Letters*, 584, 117417. <https://doi.org/10.1016/j.epsl.2022.117417>

Giant magnetofossil reviews: point-by-point responses

We would like to thank all three reviewers for their thoughtful and constructive reviews of our paper. We have endeavoured to take as many of their comments into consideration as possible.

Our responses to reviewers' comments are given in red below.

Specific changes made to the manuscript are quoted in blue (Line numbers refer to the tracked changes word document).

Reviewer 1 (Joe Kirschvink)

1. The authors provide a tour de force micromagnetic analysis of the internal magnetic structure of enigmatic giant magnetofossils that were first described by Schumann et al.[1] from sediments at the Paleocene/Eocene thermal maximum in New Jersey. They conclude that these magnetic structures could serve a role in magnetoreception, in contrast to the hypothesis that it might have been a protective structure in some microorganisms that lived in these peculiar reducing environments.

We would like to thank Joe for his positive comments on our work. It has taken a huge interdisciplinary effort to apply the 3D magnetic imaging method for the first time to a natural sample, so we are very grateful for the 'tour-de-force' comment!

2. I was initially confused about this manuscript because it is very clear that magnetite crystals that naturally have only 2% of the saturation magnetization are certainly not optimized for magnetic compass reception. That is a gross waste of iron for any organism, and iron is one of the limiting nutrients in many marine ecosystems. In the magnetotactic bacteria it is very clear that they go to great lengths to maximize the contribution to the total cellular magnetic moment from every atom of iron that they can grub from the environment (e.g., [2]). There's a huge literature on this dating back nearly 50 years[3,4]. I think the authors mean that these large magnetofossils might be more suitable for the other known type of magnetic sensing - magnetointensity reception[5]. I would suggest modifying the title of the paper to state that perhaps giant magnetofossils might play a role in the detection of magnetic intensity fluctuations in the environment, but certainly not as simple compass needles. The idea of two separate forms of magnetoreception goes back 40 years[5].

It appears our use of the term "navigational magnetoreception" in a rather general sense to refer to particles adapted to detect both direction *and* intensity has caused some unintended confusion. We did not intend to imply that the giants are optimised to act as compass needles only (indeed, we write in lines 215-217 that the moments of the giants are "more than ten times larger than the typical value of ~10 kBT for magnetotactic bacteria, and, therefore, could be considered overengineered if used solely for the purpose of magnetotaxis."). We are happy to modify this and be more specific. Joe is correct to say that the giants are not well optimised for the purposes of acting as simple compass needles – if you just want the particle to passively rotate into the direction of an Earth-strength field, then the chains of SD particles made by conventional magnetotactic bacteria are perfectly optimised for this by achieving exactly the right magnetic response with the minimum amount of Fe. We will modify the title and discussion to reflect a focus on giants being optimised for *magnetointensity* reception. We note, however, that in the proposed model, the

organism operates the receptor in a position exactly opposing the Earth's field, where the fluctuations are greatest. In this way, the organism has a way to discover the direction of the field (to within 1°) as well as measure its intensity using the same crystal (even if it is only 'optimised' to do the latter - it gets the directional information as a bonus). We cannot rule out that other receptors, specifically optimised for compass-needle response, exist elsewhere in the organism.

In view of Joe's request to change the title to refer to 'magnetointensity reception', together with Reviewer 2's request that the title should include reference to the methods used (which are, in their view, a major part of the paper), we propose the following change to the title:

"Magnetic vector tomography reveals giant magnetofossils are optimised for magnetointensity reception."

We have added citations to Joe's references 2-5 at appropriate places within the manuscript text. We have changed the abstract to refer to optimization of the giant magnetofossil for detection of intensity only:

Line 34: "Our results reveal a single magnetic vortex that displays an optimised response to spatial variations in the intensity of Earth's magnetic field."

3. In this regard, I am surprised that the authors do not mention the several instances reported by Schumann et. al[1] of clusters of these spear-like magnetofossils, apparently in an intact orientation from the original (but as yet unidentified) organism. These are clusters of up to 100 of these ~3 µm crystals, all with the spear-tip points radiating outward from a central core. These are not random aggregations of particles that have clumped together - if they were random clumps then you would expect a disordered array with the crystals pointing randomly, like a stack of needles. These structures have all of the tips pointing outwards, as demonstrated elegantly in the Schumann et al.[1] paper by using a 3 dimensional TEM tilt series (Supplemental video is on line at: https://www.pnas.org/doi/suppl/10.1073/pnas.0803634105/suppl_file/sm1.mov). Packed crystals like that could either be pointed inwards or outwards, so the chance probability of all 100 tips pointing outwards would be 2^{-100} , or $p < 1E-30$. However, the presence of many isolated giant magnetofossils that are clearly disaggregated from these clusters implies that some physical force in the environment must be able to overcome the magnetostatic binding energy between crystals in those clusters, even when in vivo. This was a puzzle for us when we first found them, as the initial TEM magnetic observation suggested they were almost single magnetic domains (but we did note that the magnetic extraction process would probably leave the particles with the transient IRM[1]). New data from this present manuscript reduces that problem substantially, as they show that the natural magnetic moments are only about 2% of the saturation value. However, I calculate that their new estimate of the magnetic moment of these particles (10^{-14} Am²) still implies a particle-particle binding energy of ~ 700 kT, assuming an interparticle separation of 1.5 µm. Hence some process must still act to disrupt the residual clumping forces between crystals in the original clusters. I think this is most easily done if they were an abrasive armament on the skin of some worm-like creature in the mud, where shear forces could disrupt them. Iron-rich dermal spicules are well known in biology and can be found disaggregated in sediments[6-8]. It is thus not implausible that these magnetofossils are some sort of spicule like the dermal granules that Lowentam and Rossman (1975) reported in the skin of Holothurians[8]. Indeed, magnetite is

biomineralized in the major lateral teeth of the Polyplacophoran mollusks (the chitons)[9], and used as a hardening agent to scrape endolithic algae from rocks. In turn, this leads to another puzzle as to how these particles could be so common in these particular dysoxic sediments. The worm (or whatever) must be fairly common to produce a high density of these objects in the sediment.

We are grateful for the reminder of these interesting features discussed by Schumann et al., which are relevant to the debate around possible functions (i.e., protection or magnetoreception). We were very careful in our paper not to attempt to debunk the protection theory in any way (it is still a possibility, despite our findings), but rather we aim to explore whether there is any evidence to support a potential biomagnetic function for the individual particles based on our experiments and simulations. For this reason, rather than detail all the evidence for the protection theory and try to knock each bit down, we cited Schuman et al. 2008 as the paper proposing the protection theory, set that to one side, and focused on exploring the magnetic structure and potential biomagnetic function. Not mentioning them at all was an oversight, and we are very happy to add reference to them in the introduction (see blue text below).

In terms of their significance, we note that we did not find anything like those 3D assemblages in any of our samples – just isolated individual crystals. We are not aware of any further literature reports of them in any subsequent papers reporting giant magnetofossils. Other researchers I have talked to who have published studies on giant magnetofossils confirm that they also did not find anything like these clusters either. These 3D assemblages are rare, therefore, and their significance is still uncertain (we note, for example, Reviewer 2's clear statement around the lack of consensus on the protection theory and their request that we maintain an open mind either way). Our results are not designed to favour one over the other, simply to explore one possibility further in light of our new insights into the internal magnetic structure of giant magnetofossils.

We have added the following reference to these features to the introduction:

Line 72: "Evidence supporting this hypothesis comes in the form of radiating clusters of up to 100 ~3 μm spearhead-like crystals, likely in an intact orientation (tip outwards) from the original organism¹. These clusters are proposed to be spicules forming the abrasive armament on the skin of some worm-like creature in the mud, where shear forces are able to overcome the magnetic forces holding the cluster together, producing some of the isolated crystals that are more typically observed. Iron-rich dermal spicules are well known in biology and can be found disaggregated in sediments²⁶⁻²⁸. Polycrystalline magnetite is also known to be biomineralised in the major lateral teeth of chitons^{29,30}, where it is used as a hardening agent to scrape endolithic algae from rocks. Clusters of giant magnetofossils have not been reported in subsequent studies, however, and none containing other types of giant magnetofossil (e.g. bullets, needles, spindles, etc.) are known."

4. None of the above reasoning, however, argues against the conclusions in the present manuscript that novel particles related to these giant magnetofossils might actually be evolved in magnetointensity detection. That is an elegant concept, even if it is unlikely in the organism that left these particular fossils. Small worms in this environment would have been unlikely to be migratory, and so probably would not needed to have a magnetointensity receptor in an environment where spatial magnetic fields don't change much and little oxygen is available. My reasoning is as follows: In the initial suggestion that biogenic magnetite might be involved in the intensity component of magnetoreception, Mike Walker and I [5] developed the

“thermally driven variance model of magnetointensity reception”. This simply demonstrated that neural signals from magnetosomes attached to mechanoreceptors could convey both directional information and intensity information to an animal’s brain. The trick there was to realize that strong fields damp Brownian motion by increasing the magnetic to thermal energy ratio ($\mu B/kT$), pinning the moment to the background field direction, whereas thermal agitation and large angular motions dominate in weak fields. Hence, the variability in the action potentials firing rate from these cells is a monitor of total field sensitivity, whereas the mean firing rate would yield compass information. By analyzing the spatial derivatives of the Langevin function, we [5] concluded that the optimum sensitivity to magnetic intensity fluctuations would be at relatively small SD crystals, with $\mu B/kT$ ratios of about 2. Winklhofer and I [10] updated this analysis to consider in the context of mechanical torques that might be needed to produce such effects, but the basic principle is the same. Thus, the suggestion in this manuscript that the extraordinarily high magnetic susceptibility estimated for these giant magnetofossils could offer a different magnetosome-based path to magnetointensity reception is a novel idea. I would think that once these things evolved in some small worm, it might have been exapted for a component of magnetointensity reception in a more migratory descendent.

We have added citations to Joe’s reference [5] as the original model for magnetointensity reception based on thermally driven variance (e.g. line 297). The idea of exaptation is a fascinating one, which enables both protection and magnetointensity hypotheses to be true at the same time, rather than being mutually exclusive! We have added reference to the idea in the discussion:

Line 441: “Even in the case that the particle studied here evolved for the purposes of protection in some small worm, the possibility remains that it might have been exapted for a component of magnetointensity reception in a more migratory descendent.”

5. In summary, I recommend publication of this important study following revisions along the above thread, and in response to the comments below.

We thank Joe for his thoughts and ideas on this, and hope that our changes reflect the discussion, within the constraints posed by Reviewer 2 who wishes us to remain strictly on the fence with regard to favouring one theory over the other.

More detailed comments on text:

6. Line 78: It is *not* optimized for detecting magnetic field direction. Iron is one of the principal limiting nutrients in biology, and building a compass needle that is only 2% efficient is a waste. I strongly recommend removing these misleading statements throughout the ms.

Yes we agree. We have changed this line (and throughout where necessary) to:

Line 104: “Armed with this knowledge, we calculate its magnetoreceptive response using a torque-transducer model6 and demonstrate its optimised potential to sense the Earth’s magnetic field intensity.”

7. Lines 134-135: Could the Bloch Point singularity described here be caused by a spiral screw dislocation in the magnetite crystal lattice? A spiral screw dislocation

along the particle length would help the biomineralization process, but it has never been documented in HRTEM analyses of bacterial magnetite. Does your data on the natural crystals rule this out?

Our data do not rule this in or out. However, we note that Bloch points are reproduced in our micromagnetic simulations without the need of a dislocation. They form in response to a magnetic field of the order 10 mT applied antiparallel to the core magnetization. They also formed naturally as part of a nudged elastic band calculation of the energy barrier to reverse core magnetisation.

8. Lines 193-196, and at the end of the supplemental information. I am not surprised that the vortex geometry of these crystals would produce such a high intrinsic magnetic susceptibility. However, the analysis would be much more powerful if you calculated both the real and imaginary components of the complex magnetic susceptibility for the giant magnetofossils and other particles and add them to Table S2. That would help us compare the intrinsic properties to past studies of superparamagnetic magnetite[11], for example. It would also contribute greatly if you could calculate the imaginary component (χ'' , that controls energy dissipation) out to a few GHz. These particles might have applications in the ferrite industry, and such calculations might lead to inorganic synthesis and potential industrial applications. I'm not familiar enough with the MERRILL software to know if that is feasible.

We are unable to calculate the imaginary part using the MERRILL software, as this code does not account for any time-dependence of the magnetisation. We looked into the possibility of doing this using other code, but were advised that this would be a major undertaking and very time consuming (weeks/months). We consider this to be out of the scope of the present manuscript but will bear it in mind for future studies.

9. Line 205-206, Again, this is an inefficient use of metabolic iron.

We now make this point when stating that the particle is overengineered for the purposes of magnetotaxis:

Line 287: "A value of $MB = 121$ kBT is found for the giant spearhead, assuming a representative Earth field strength of $50 \mu\text{T}$ and a temperature of 298 K. This is more than ten times larger than the typical value of ~ 10 kBT for magnetotactic bacteria³², and, therefore, could be considered overengineered (and an inefficient use of metabolic iron) if used solely for the purpose of magnetotaxis."

9. Lines 220-221: "The eukaryotic cells are interesting in bridging the gap between magnetotactic bacteria ... and origin in Eukaryotes." Hey, this is precisely the proposal that Vali and I[12] made in 1991. (A PDF is on my www site if you don't have the book.)

We have added a citation to Vali and Kirschvink.

10. Lines 229 to 238: I have a few comments here. First, direct extracts followed by TEM analysis of the olfactory tissues of trout and salmon (genus *Oncorhynchus*) actually

revealed magnetosome chains which were very similar to those from the magnetotactic bacteria, including {111} crystal twins (see Mann et al., 1988[13]). The MFM technique of Diebel et al.[14] (2000, ref #45) identified clusters, but could only map out the minimum size of the magnetite-containing cells, as the attraction of the probe dropped off as $1/r^4$. There was no 3-D information in that analysis. Hence, the best data for the moment of these structures is from the second study you cite, Eder et al. (2012, ref. #46), where the whole cells were rotating. There is no evidence for two size categories of the magnetite crystals, and the TEM analysis of Mann et al.[13] of the olfactory extracts from these tissues most certainly would have identified the giant magnetosomes if they had been present in the olfactory epithelium. The 1997 paper by Walker et al.[15] demonstrated that magnetite-containing cells in that tissue were connected to magnetically-responsive nerve fibers in the supraorbital trunk, using a lipophilic dye, so it is clear that they are magnetoreceptor cells. Eder et al.[16] even showed that rotating the magnetic field on stabilized cells caused Ca^{++} ion channels to open. Note that the Edelman et al. (2015)[17] attempt to replicate the Eder et al.[16] rotating cell paper should just be ignored, as they failed to follow well-known protocols for removing contamination in the reagents and chemicals used for disaggregating the trout olfactory epithelium. They apparently had not seen the 1995 Nature paper by Kobayashi et al.[18] showing that such contaminants were ubiquitous in common 'reagent-grade' chemicals and solutions. It is therefore no surprise that Edelman et al. only found contaminants.

We have made this point and cited the Kobayashi paper:

Line 318: "Although these results were later challenged as being due to extracellular contamination⁵⁴, that result may itself be due to failure to remove contamination in the reagents and chemicals used for disaggregating the trout olfactory epithelium⁵⁵. Structures similar to those originally described by ref 52 have since been confirmed by clean-lab extraction of magnetic cells from the olfactory epithelium of salmon⁴⁶. "

11. My conclusion is that if giant magnetosomes like the one Harrison et al. have modeled are part of the magnetointensity receptor complex in higher animals, they are not co-located with the compass receptor cells. Finding them is a worthwhile goal.

As discussed above, we agree the particles studied appear to be optimised for measuring intensity. We cannot rule out the presence of other magnetoreceptors in the animal that are optimised more specifically for just direction, but note that these particles do deliver good directional information as a useful bonus, so separate magnetoreceptors are not *required*.

12. Line 258 – again, Ref. #45 is an under-estimate.

We recognise this and state in line 341 that this value is considered "the minimum baseline case".

13. Lines 269-270. The nervous system is very good at integrating small signals from a large number of receptor cells to extract a small signal from the background and doing pattern recognition; this is how a parent can pick up the tiny, faint sounds of their child at day care against the background of all of the other children (at least I could). Thus, if one large spearhead could be accurate to 1° with a mass 300x that of a 6 kT magnetosome chain of accuracy 10° , I would say that an array of 300 of the 6 kT chains would give the brain an accuracy of $10/\sqrt{300}$, or 0.56° . The analysis presented here does not make sense in terms of how the nervous system works.

Again we are happy to concede that these are not primarily optimised for direction. We have added the following to the end of that discussion:

Line 350: “Hence, although the spearhead is primarily optimised for magnetointensity reception, it also delivers an accurate means to determine the field direction. We cannot rule out the presence of other magnetoreceptors in the animal that are optimised more specifically for just direction.”

14. Lines 272-307, and Fig. 4. This is definitely a variant of the ‘thermally-driven variance model’ of magnetoreception⁵, and should be cited as such. The difference is that you include a rigidity of the attachment fibers that changes the tuning of the system. However, I am still not convinced that there would be an advantage of a crystal of magnetite of only 2% saturation, compared with an assemblage of smaller crystals which could achieve the same magnetic moment with correspondingly less iron required (see above). You should also go into the literature in neurobiology concerning the two major categories of mechanical neuroreceptor cells – phasic and tonic (google it!). The well-known difference lends itself nicely to the understanding of the two forms of magnetoreception – phasic cells could easily determine a compass direction, whereas the tonic cells could monitor the intensity.

We now cite the thermally driven variance model that preceded the more advanced model used in this paper.

We recognise and consider the alternative strategy in the text and put forward some ideas as to why there may be advantages to the single crystal approach. Line 391: “Beyond the single particle scenario, the alternative strategy to increasing moment is by building ever larger clusters of uniformly magnetised magnetite nanoparticles³⁰. However, this strategy may come at the expense of the domain state predictability offered by a single, giant magnetofossil, where features such as vortex cores form in a natural and predictable way due to the interplay of shape anisotropy and magnetocrystalline anisotropy. In addition, the well-defined crystal terminations of giant magnetosomes (Fig. 1a) could provide a more controllable and reproducible means of anchoring the particle within the cell.”

15. Lines 347-348. Ah, the presence of functional magnetoreception in extant mollusks, amphibians, fish, reptiles, birds, and mammals (among other groups) argues that magnetoreception dates back well before the Cambrian explosion (to at least the first bilaterians 550 Myr).

Our aim here was to point out here that there is a way to find direct *fossil* evidence for it. This could drive the desire to search for older giants and discover when they first appear. We have included Joe’s statement to the effect:

Line 460: “Although the presence of functional magnetoreception in extant mollusks, amphibians, fish, reptiles, birds, and mammals (among other groups) argues that magnetoreception dates back well before the Cambrian explosion (to at least the first bilaterians 550 Myr), if confirmed, our work would provide the first *fossil* evidence that navigational magnetoreception developed as a sense in eukaryotes at least 97 million years ago.”

I look forward to seeing a revision!

J.L. Kirschvink

Reviewer 2

1. This manuscript has a lot to it, and I am impressed the authors were able to parse all this information into a compact manuscript and story. The authors show it is possible to directly measure and visualize the internal magnetic structure of giant magnetofossils. This is novel and impactful. This is really neat. Their method has potential for groups studying giant magnetofossils, Early Earth, and life on other planets.

We are grateful for the comment recognising the novel and impactful nature of the work, and its potential broader use. We hope to apply our work to those fields in the near future.

2. Unfortunately, as written, I think this message gets muddled and overshadowed by some of the interpretations being made about giant magnetofossils as a whole (especially since this manuscript focuses on a single spearhead and does not look in detail at other morphologies). With some restructuring and focusing of the manuscript toward the authors' method (which I think is the most impactful part of the manuscript), and the subsequent importance of it with proposed interpretations separated, I think it may be suitable for publication in Communications Earth and Environment.

We put a lot of thought into considering whether to fully restructure the paper along the lines suggested by reviewer 2. In doing so we took the following aspects into account. First, although the methods used are applied here for the first time to a natural sample, they are not inherently new and have been discussed in the physics literature previously. It is therefore not appropriate to structure a paper in Nat Comms Earth and Environment entirely around the technical aspects. Second, neither of the other two reviewers raise the focus of the paper as being an issue (quite the opposite - reviewer 1 engages fully with the topic and its importance, and reviewer 3 praises the introduction as "great for a general audience"). In our view, therefore, the suggestion to refocus the paper on the methods would make it far less suitable for the Nature Comms Earth and Environment audience.

Having said that, we do agree with the reviewer that the methods presented here for the first time applied to natural samples are going to be impactful on the broader field of rock and paleomagnetism. In light of that we have modified the title of the paper (see response to reviewer 1) to include direct mention of the method used. We have included an opening paragraph to the Results section that highlights the method and its broader impact:

Line 112: "The sample studied is a "no-stalk" spearhead extracted from a ~56 million year old pelagic marine sediment from the J-Anomaly Ridge, North Atlantic¹⁵ (Fig. 1a). The particle has a diameter of 1.1 μm and a length of 2.25 μm , and comprises an approximately cylindrical base and a cone-shaped tip (Fig. 1b). The dimensions of the particle make it highly absorbing to resonant soft X-rays and electrons, which poses a major challenge to probing its internal magnetic structure without destructive sampling (which would irreversibly change its magnetic state). Most transmission-based nanomagnetic imaging methods, such as electron holography³⁴ and scanning transmission X-ray microscopy³⁵, are limited to samples thinner than ~300 nm. Recent technical breakthroughs, however, open up the opportunity to image natural

Fe-oxide samples in the multi-micron size range^{4,5,36}. Soft X-ray pre-edge dichroic ptychography³⁶ works by tuning soft X-rays to energies far below the Fe-absorption edge, enabling them to pass through much larger samples. Magnetic vector tomography^{4,5} then enables all three components of magnetisation to be reconstructed and spatially resolved throughout the volume of the grain with a resolution of a few 10s of nm. This combination of techniques opens up the entire single- to multi-vortex size range of natural remanence carriers to 3D magnetic imaging. Crucially, these breakthroughs mean that the magnetic information accessible to experimental observation matches precisely that which can be accessed through micromagnetic simulation. It is now possible, therefore, to test and verify the predictions of micromagnetic theory by direct comparison of predicted versus observed behaviour in 3D at the individual grain scale – a capability that will impact the broader fields of rock magnetism, paleomagnetism and environmental magnetism.”

Comments and requested revisions:

3. I think this method and investigation of giant magnetofossils is significant and impactful. However, I was put off by the overall tone and claims from the manuscript. I don't think you meant to do this, but there seems to be a lot of overarching, sweeping statements about all giant magnetofossils in general (see line-by-line comments below). I encourage you to focus on the cool thing about your work, the method and findings, and less-so about making these huge statements until making the end of the manuscript when you make interpretations. Also, please make it more clear that many of these statements and interpretations are on giant spearheads, based on this study, before extending them to other giant magnetofossil morphologies. I recommend really highlighting what makes your paper so distinct from other papers that have discussed the size and domain state of spearheads before. It would also be helpful if you included headers and split the manuscript into more distinct themes (e.g., “Modeling the magnetic behavior” vs “Explanation for single-particle magnetoreception”...)

See above for inclusion of paragraph highlighting the novelty of our work.

We have included section headings to break up the paper.

We have checked the text and removed reference to ‘all magnetofossils’ (e.g. in the abstract) to make it clear that our *experimental* observations pertain only to “the giant spearhead” studied. Later when discussing *simulation* results we are very clear that these simulation take into account other morphologies, e.g.:

Line 359: “To address whether this optimisation in magnetoreception performance can be generalised to other giant magnetofossils, the field sensitivity of $\Delta\psi$ and the minimum detectable deviation angle at the critical point are plotted in Fig. 4c for a range of biologically plausible values of K . The calculated moments of 24 giant magnetofossils (Fig. S6) with a range of volumes and morphologies (including needles, bullets, kinked bullets, spearheads and spindles) are presented in Table S1 and as a histogram in Fig. 4c.”

4. Again, this method combination is interesting, unique, and envelope-pushing. It is exciting to learn and read about. I am not an expert in these measurements and techniques, but I am a little concerned about the reproducibility of these datasets.

We are happy to reassure the reviewer that, although we have performed the very

first analysis of a natural iron oxide sample here, the methods themselves are well established in the physics and nanomagnetism communities. Each 3D dataset is results of ~200 projections, which produce reproducible and consistent data. The features of the 3D reconstruction can be verified against any single projection, which are all self consistent with the presented 3D structure. We have collected multiple 2D and one subsequent 3D dataset using a different synchrotron beam line (although not yet processed and reconstructed) on a range of different giant magnetofossil morphologies (including spearheads, bullets and needles). The datasets are highly reproducible (see similar example below, displaying a vortex structure). Initial indications from the 2D projections we have are that the method works reliably and reproducibly. This work will form part of a larger follow up paper.

Figure shows an SEM image of different no-stalk spearhead with the inset showing the XMCD phase image. Red and blue are magnetisation into and out of the plane, reproducing the same vortex structure observed in Fig. 1 of the manuscript.

5. Only one spearhead was directly measured and that does not represent natural variability between spearheads and other giant magnetofossil morphologies. For example, you do not fully consider that many spearheads are found with stalks attached to them, which is sure to effect the magnetic characteristics of the spearhead too (see your supplemental figure 6 with simulations). To this point, there

is actually a wide range of proposed magnetic domain states of spearheads due to the varying sizes (Wagner et al., 2024; Xue, Chang, Pei, et al., 2022; Xue & Chang, 2024). Please bring in more discussion about the spearhead variation and caveats associated with this. For example, what role do you think the stalks play/how do you think they would influence the magnetic domain state of the spearheads? In future work I would implore you to strengthen this study with replications on other spearhead morphologies (e.g., more/less elongated and with/without stalks), other giant magnetofossils morphologies, and from different time periods and sediments to account for natural variability.

From a simulation perspective, we consider a broad range of morphologies in the Supplemental Information and Fig. 4 in the main text and discussion. For example, the cluster of remanent moments around $MB = 100$ kBT labelled as spearheads in Fig. 4 includes both stalk and no-stalk varieties: there does not appear to be any major difference between the two in terms of moment. The simulated remanence states for the stalk varieties presented in the supplement show that the stalk contains the vortex core, and that the core then continues through the rest of the particle. The advantage of a stalk may be that it provides a way to anchor the vortex core in the centre of the base of the particle, which could add some extra magnetic stability. However we have demonstrated here that the vortex core is already stable enough in the no-stalk variety, so this may not be that significant.

6. The spearhead studied here was magnetically extracted from the sediment such that it is not in its natural magnetic state of “life” position. As noted in the manuscript, it is also surrounded by other detrital particles which were magnetically extracted and have magnetization too. You also note that the spearhead has a high susceptibility. All these things make me concerned about whether you are capturing the “true” magnetic state of the spearhead. Please address these concerns within the manuscript with caveats or supportive reasoning. As another aside, how do you think having such a high susceptibility (especially compared to MTB) is beneficial for magnetotaxis? How does this compare to organisms that we know make magnetic particles, but not for magnetoreception (e.g., dissimilatory reducing bacteria, sclerite from gastropods)? Comparing to these organisms could help bolster your arguments.

As stated in the manuscript (line 234 “These results suggest that the Bloch point and tip domain are likely caused by exposure of the sample to a magnetic field during magnetic extraction from the sediment (see Methods), rather than being primary features of the fossil’s natural, lowest energy magnetic state.”) we believe that the experimentally observed magnetic structure has indeed been modified from its true magnetic state by the formation of the Bloch point and the tip domain under the influence of the magnetic field applied during extraction. Our simulations, however, support the stated conclusion that the vortex structure and medial domain wall are part of the true magnetic state prior to magnetic extraction. There are four lines of evidence to support this: 1) the simulated structure shown in Fig. 3 is the lowest energy remanent state of the particle; 2) the same structure is obtained as the saturation isothermal remanence structure in hysteresis simulations for fields applied along the long axis; 3) the same state is obtained using the layer-by-layer growth models from the base up and the tip down; and 4) the Bloch point can be formed from the simulated structure shown in Fig. 3 by applying a field of a few mT, which is well within the range of fields involved in magnetic extraction. We remain confident in the conclusions as stated.

The role of susceptibility in magnetoreception is discussed quantitatively in the Supplemental Information (Equation S4 and S5). The high anisotropic susceptibility of the giant magnetofossil creates an additional torque on the particle for fields

applied at an angle to the long axis (and a small increase in absolute moment for field applied parallel to the long axis). The magnitude of the effect is calculated to be of the order 1% of the torque produced on account of the large remanence moment of the fossil. We conclude, therefore, that the role susceptibility is secondary to that of the absolute remanence of the fossil.

7. Another note on reproducibility: this method seems complicated. Will other researchers, like the ones you discuss in the manuscript (paleontologists, etc.), really be able to reproduce and easily implement this method to directly measure giant magnetofossils? What will it take to make it happen?

Although there are several synchrotron beamlines that are able to perform such measurements, the data acquisition and processing are currently both labour and time intensive, and do require a high level of expertise and experience (hence the broad range and large number of co-authors for this collaboration, which includes many beam line scientists as well as the pioneers of 3D magnetic imaging from the physics community!). However, these are all surmountable barriers of a practical nature that will reduce quickly over time. Having performed this first demonstration of the methods applied to natural samples, we aim to eliminate these barriers by establishing standard experimental protocols and reconstruction codes that can be made generally available.

8. My first thought when you started to describe the Bloch point was that it was going to be a descriptive feature of magnetofossils. I worried about bias from magnetic extraction, but then you addressed that. I'm a bit confused why there is so much emphasis on the Bloch point in the manuscript then, especially since you state that it's unlikely a natural signature? I think better/more clear organization of the manuscript could help clear this up. For example, if you are simply trying to describe the magnetic characteristics you identified, consider putting all of this into a section called "Identifying magnetic characteristics from direct measurements and simulations" or something.

This is, to our knowledge, the first observation of a Bloch point in the vortex state of magnetite and in any natural sample. In more general terms the presence of a Bloch point is highly significant in terms of understanding the mechanism of how vortex states are able to reverse their magnetisation. For example, from a paleomagnetic perspective it is critical to understand the mechanism of moment reversal in vortex states, which are common carriers of paleomagnetic remanence – by knowing the mechanism we can better constrain the activation energy barriers involved in the nucleation, propagation and annihilation of Bloch points, thereby better understanding the acquisition and stability of paleomagnetic remanence.

9. This is picky, but the term "magnetosome" refers to the entire organelle that encompasses the magnetic particles. In general, please only use magnetosome when referring to the entire organelle. This will satisfy any MTB microbiologists reading your paper.

Thanks for the clarification. We have adjusted our use of the term magnetosome accordingly throughout and refer simply instead to 'magnetic particle' or similar.

10. Lines 24-27: This comes off very strong, like we have a complete consensus that giant magnetofossil are biogenic. I agree that they are likely biogenic, but I encourage you to make this more conservative. Until we find the organism responsible for making them, we cannot say for certain.

We have added a caveat to the statement along the lines requested:

Line 63: “Their widespread occurrence, chemical purity, oxygen isotopes, crystallographic perfection, crystallographic orientations, allometric relations and distinctive morphologies (comprising needles, spindles, bullets, kinked bullets, socks and spearheads^{2,2}, and potentially also seeds, squash and spades³) provide compelling evidence of biogenicity, although until the organism responsible for making them is identified, this evidence remains circumstantial.”

We have added reference to Pei et al. (2025) which provides new 3D morphological varieties (kinked bullet and sock) and further evidence in support of a biogenic origin. We have also pointed out that finding evidence of biomagnetic optimisation adds to the evidence supporting a biogenic origin:

Line 93: “Finding such evidence of biomagnetic optimisation would add to the already substantial body of evidence supporting a biogenic origin of giant magnetofossils.”

11. Lines 27-30: This is also written very strongly, like there is a consensus between everyone that the main function of giant magnetofossils is for protection, but that is not true (Wagner, Egli, et al., 2021; Wagner, Lascu, et al., 2021; Xue, Chang, Dickens, et al., 2022; Xue & Chang, 2024). Other groups have suggested this “alternative” hypothesis already, that they are used for magnetoreception (see the same references as examples). Please change the wording in these sentences to better reflect the variety of interpretations already put out there, for e.g., “The most common interpretation is that ...” “We provide more evidence in favor of the hypothesis that that giant magnetofossil producing organisms exploited...”

We have added these citations to the suggestion of a magnetoreceptive function:

Line 81: “An alternative hypothesis, considering their morphological similarity to some conventional magnetofossils, is that these crystals were exploited to perform some kind of biomagnetic function^{2,15,16,19,20}.”

12. Lines 31-34: Do “all” giant magnetofossils show the same behaviors/magnetic traits? I think you mean that to say this for spearhead-shaped fossils only.

We have removed reference to “all” giant magnetofossils in the abstract. Other types are discussed later in the paper.

13. Line 36: Why just eukaryotes? There are prokaryotes large enough to “house” them. Again, I have some issues with these strong statements since we do not know what organism made them. Please add another brief sentence here to make the connection between why these magnetic traits and navigational magnetoreception earlier in the fossil record.

We have removed “eukaryotes” in the abstract. Later in the introduction when discussing this we cite the original Schuman et al. paper who argue for eukaryotes based on the large size of the particles:

Line 72: “Due to their large size, giant magnetofossils are not considered to be the product of bacteria, but rather (as yet unidentified) eukaryotes¹.”

14. Line 38: iron oxide

Added.

15. Lines 44-47: Reword or break into two separate sentences

We have split into two sentences:

Line 48: "Giant magnetofossils were initially thought to be exclusively associated with hyperthermal events, and therefore considered useful indicators of past climates and environments. However, they have since been identified globally in sediments both modern and ancient and from periods of global cooling as well as warming^{3,15–21}."

16. Lines 48-51: Please add the corresponding references for each of these evidences in your sentence. Also, what about iron isotopes, growth structure, and magnetic signatures (Amor et al., 2016, 2018; Mathon et al., 2024; Wagner, Egli, et al., 2021; Xue, Chang, Pei, et al., 2022)?

Have added citations to the list. As far as we know the work of Amor and Mathon pertains only to conventional MTB, not giants. The work of Wagner et al. 2021 pertains to detection of giant needles due to their magnetic signature, not to the question of abiotic vs biogenic origin. We have added references to Xue et al. 2022 and Pei et al. 2025.

17. Lines 55-60: Do you mean to say that this is just with respect to spearheads? Otherwise, I have similar comments to those above about the abstract. Please reword this to better reflect the variety of interpretations out there for all giant magnetofossils. There is no general consensus and a lot of people already agree that they were used for magnetoreception. Also, most people say that the organism(s) responsible are likely eukaryotes because of the size of giant magnetofossils, but it's still not impossible that they are/were prokaryotes.

See above.

18. Lines 60-65: Break this up into a couple of sentences.

Done.

19. Lines 70-75: Micromagnetic simulations were performed on giant needles imaged using TEM and their corresponding FORC signatures were modeled from this too (Wagner, Egli, et al., 2021). Please clarify that magnetic studies haven't been done directly on giant spearheads, which are the main focus of your study.

Added reference to Wagner et al. 2021 in list of micromagnetic studies.

20. Line 75-76: This is the really cool, important, and new thing that your manuscript brings to the table: a way to directly look at the internal magnetic structure of a giant

magnetofossil. I recommend emphasizing and focusing on this as your main takeaway.

See above for a new paragraph highlighting the broader impact of the method.

21. Lines 80-85: Please include a brief sentence of how you obtained the particle. Did you magnetically extract it from the sediment? Why this one? There's a disconnect because the methods section is separated in this format.

Yes the particle was magnetically extracted (as described in detail in the method section). We have moved the method section to come before the references, as per journal guidelines. In terms of why this particle was chosen – it was the one closest to the centre of the grid and therefore least shadowed by the holder when tilting. We have added a statement to this effect to the methods:

Line 502: "Several samples were prepared with the aim of finding a representative fossil close to the centre of the Si₃N₄ membrane, which facilitates data acquisition. The position of the particle on the membrane was identified using SEM."

22. Lines 187-227: In addition to comparing to magnetite biomineralized by MTB, which we know perform magnetotaxis, I recommend adding some comparisons to the magnetic properties of iron particles made by other organisms for purposes other than magnetotaxis and magnetoreception (e.g., protection, which you are arguing against in this paper), as negative supporting arguments (e.g., Fe-reducing bacteria that facilitate magnetite precipitation, the scaly-foot snail, etc.)

We have added the following sentence referring to the quite distinct properties of magnetite produced by other organisms for non-magnetic purposes:

Line 269: "In terms of potential non-magnetoreceptive applications, it is worth noting that the observed microstructural and magnetic characteristics of the giant spearhead magnetofossil are very different to those of magnetite produced by other organisms for non-magnetic purposes (e.g., the magnetite nanorod/chitin fibre composite structure produced for hardening/abrasion purposes by chiton³⁰ or the ultrafine-grained magnetite particles produced as a metabolic byproduct by dissimilatory iron-reducing bacteria⁵¹)."

23. Line 190: consider a brief sentence relating to the remnant moment of MTB magnetite chains

A comparison of remanence moment of the giant vs MTB is provided a little further down (Line 289).

24. Line 191: ...coercivity $B_c \sim 2\text{mT}$ (40 times greater than the Earth's field) and coercivity..." Line 206: replace magnetosomes with something else... "magnetic particles made by MTB"

Done.

25. Lines 215-219: This (their remanent moment) seems the most compelling evidence to me that they could have been used by another, larger organism like the ones you compare them to (MMPs and eukaryotes). I recommend comparing this to abiotic magnetite crystals to be safe/strengthen your biogenic argument

We do this in line 280 and Figure S5.

26. Line 219: I think you mean magnetic particles as “magnetosome” refers to the entire organelle

Done.

27. Line 226 and 239: magnetic particle, not magnetosome

Done.

28. Line 252: this part about weaker field strengths seems to be something you can test

29. Lines 291-295: How do these magnetic energies relate to magnetite chains made by MTB and the protists and eukaryotes that we know use magnetoreception in the modern?

They are much larger - and we argue that this larger moment is good for intensity detection. Observed magnetoreceptors in eukaryotes are discussed - they are usually not large single crystals like these, but we do discuss the potential advantage that being a single crystal might provide (e.g. better anchoring, structural rigidity). The large moments of other fossil morphologies are presented in Fig. 4c.

30. Line 296-298: I agree that whoever made them produced them through controlled biomineralization to prevent them being too big or small, but be careful because there are some reported larger than 3 microns (Xue, Chang, Pei, et al., 2022)

Have changed to 3-4 μm to be consistent with Xue et al. 2022 (the VAST majority are below 3 μm but some do reach 4 μm).

31. Lines 299: How are you predicting the domain state stability for all giant magnetofossils? Is this based on your magnetic energy calculations or is this just with respect to giant spearheads?

Micromagnetic simulations for different morphologies are presented in the Supplemental (based on the 3D tomography results of Pei et al. 2025).

32. Line 303: What about the role of the stalks? Could those be helpful in “anchoring”?

Yes - see simulations and discussion above.

33. Line 303 and 305: please use magnetofossils or crystals, not magnetosomes

Done.

34. Lines 341-345: Please be careful with this statement. Previous studies have shown that there are actually very few giant magnetofossils and it can be difficult to find them at all (Kadam et al., 2024; Wagner et al., 2024; Wagner, Lascu, et al., 2021)

We have changed this statement to:

Line 455: "Whatever organism is responsible, they should be sufficiently abundant to explain the widespread occurrence of their fossilised remains in sediments (albeit at several orders of magnitude lower concentration than conventional magnetofossils), which would favour smaller culprits over larger ones."

35. Line 348: References: Wagner et al., 2024; Xue & Chang, 2024

Added.

36. Line 517: Why/how did you choose this one spearhead? (besides it being in the center of membrane)

We chose this spearhead indeed because it was at the centre of the membrane, which makes data acquisition possible as it was the least shadowed by the holder when tilting. It also was large enough to test the limits of what can be achieved in terms of sample size.

37. Supplemental Figure S6: I recommend blending in more of the spindle variety simulations into your main text discussion about optimization of your measured spindle with respect to different lengths/widths and stalks

We have performed 4 additional simulations with spindle geometries estimated from 2D TEM data. The results have been added to the Supplemental Information (Table S1 and Fig. S6), and are also now included in the histogram of magnetic energies in the main Figure 4. Interestingly, the spindles with [110] long axis have single vortex states with some of the highest magnetic energies (up 530 kBT), which adds further weight to the overall hypothesis we are putting forward here, with the mean magnetic energy now being 93 kBT, close the 100 kBT cluster already discussed. We have added references to the spindle values to the text.

Line 361: "The calculated moments of 24 giant magnetofossils (Fig. S6) with a range of volumes and morphologies (including needles, bullets, kinked bullets, spearheads and spindles) are presented in Table S1 and as a histogram in Fig. 4c. The mean magnetic energy is 93 kBT, with a median value of 36 kBT. The maximum value is 523 kBT (for a spindle) and values cluster around 30 kBT for needles and giant bullets and around 100 kBT for spearheads. The two largest values both correspond to spindles."

Line 384: "The clustering of magnetic energy values for spearheads around $\sim 100 k_B T$ might, therefore, be considered as striking the 'optimum' balance between improved performance and domain-state predictability (and, notably, is close to the mean value

of all fossil types modelled here). Support for this principle of optimisation is found in the size distributions of giant magnetofossils¹⁵. For example, the remanent moment of needles increases by increasing their overall length whilst maintaining the very high aspect ratio needed to stabilise the uniformly magnetised state⁶². Once needles have reached an average diameter of 100 nm, they are observed to grow in length only¹⁴, reaching values up to 2 μm (corresponding to a magnetic energy of $92 k_B T$ – again, similar to the mean of all fossil types).”

Reviewer 3

1. The authors leverage magnetic vector field tomography to visualize the magnetization in a giant spearhead magnetofossil and use an analytical equation to assess its magneto-receptive response. The introduction is great for a general audience; the results may be of interest to the readership of *Communications Earth & Environment* following major revision.

This is nice to hear!

2. Clearly state the energy used for pre-edge XMCD measurements (and shift from resonance) at the beginning and also include in caption. Please add details for “XMCD projections acquired about two rotation axes”, e.g., y and z axis with x-rays incident along x axis.

We have added these details to the caption (already in the methods section). Details on rotation axes and X-ray beam direction are in the caption.

3. The authors show the crystallographic orientation and state magnetostriction values. How is the crystal orientation determined? I couldn't find any data on this. How is the magnetostriction considered in both experiment and simulation?

We did not measure direction on this exact sample due to sample membrane failure, however the spearhead shape is very reproducible and other essentially identical crystals were found. Electron diffraction was performed on other no-stalk spearheads, showing that they have their long axis oriented along 113 (now published as Pei et al. 2025). We performed simulations with a range of orientations and found 113 fits best. We did not take magnetostriction into account (an earlier draft of the paper performed some simulations under uniaxial compression to test the effect on the Bloch point, but these simulations no longer appear as they are out of scope – we have removed mention of the magnetostriction constant).

4. Taking into account the authors' statement “These results suggest that the Bloch point and tip domain are likely caused by exposure of the sample to a magnetic field during magnetic extraction from the sediment (see Methods), rather than being primary features of the fossil's natural, lowest energy magnetic state.”, how can they be sure that not the entire magnetization configuration has been altered? If this was a likely scenario, would it somewhat affect their conclusions/key outcomes?

See response to Reviewer 2 above, repeated here for convenience:

As stated in the manuscript (line 234 “These results suggest that the Bloch point and tip domain are likely caused by exposure of the sample to a magnetic field during magnetic extraction from the sediment (see Methods), rather than being primary

features of the fossil's natural, lowest energy magnetic state.”) we believe that the experimentally observed magnetic structure HAS indeed been modified from its “true” magnetic state by the formation of the Bloch point and the tip domain under the influence of the magnetic field applied during extraction. Our simulations, however, support the stated conclusion that the vortex structure and medial domain wall ARE part of the “true” magnetic state prior to magnetic extraction. There are four lines of evidence to support this: 1) the simulated structure shown in Fig. 3 is the lowest energy remanent state of the particle; 2) the same structure is obtained as the saturation isothermal remanence structure in hysteresis simulations for fields applied along the long axis; 3) the same state is obtained using the layer-by-layer growth models from the base up and the tip down; and 4) the Bloch point can be formed from the simulated structure shown in Fig. 3 by applying a field of a few mT, which is well within the range of fields involved in magnetic extraction. We remain confident in the conclusions as stated.

5. What magnetization value M was used as pertaining to “In the context of magnetotaxis, the alignment efficiency can be quantified by the ratio of magnetic to thermal energy (MB/kBT , where M is the magnetic moment in Am^2 , B is the Earth's field in Tesla, kB is Boltzmann's constant and T is temperature).” Was this the saturation magnetization of the remanent magnetization or the net moment of the elongated vortex core or the adjusted magnetization at the applied magnetic field as determined from hysteresis loops? It can surely not be the former.

We can confirm that it is the remanent moment used, not the saturation moment. We have added the word remanent to the text to make this clear.

6. Given the S-shape of the hysteresis compared with the square shape of nanocubes, it is not surprising to obtain a higher susceptibility for the former. However, we would that be beneficial? A rigid moment should be superior in terms of magnetoreception.

See response to reviewer 2, repeated below for convenience:

The role of susceptibility in magnetoreception is discussed quantitatively in the Supplemental Information (Equation S4 and S5). The high anisotropic susceptibility of the giant magnetofossil creates an additional torque on the particle for fields applied at an angle to the long axis (and a small increase in absolute moment for field applied parallel to the long axis). The magnitude of the effect is calculated to be of the order 1% of the torque produced on account of the large remanence moment of the fossil. We conclude, therefore, that the role susceptibility is secondary to that of the absolute remanence of the fossil.

7. The authors compare their magnetic state with literature: “The dominant feature is a single vortex with curved/kinked core trajectory (Fig. 2a), rather than the multi-domain state previously predicted for slightly larger spearheads with a stalk^{2,14} or the uniform state previously claimed on the basis of 2D electron holography observations¹.” While this is appropriate I recommend rephrasing. Clearly the samples must have been different. There is no way electron holography would be misinterpreted that strongly. To this extent, the results should be presented as corroboration of highly sensitivity of shape (or similar, please elaborate).

The Schuman paper does indeed state single-domain on the basis of their interpretation of spearhead electron holography (and Reviewer 1 mentions that again in their review here, but also suspecting that it may have been affected by magnetic extraction). We are sceptical of that interpretation given the figure in their paper, and suspect there may have been a failure to properly account or correct for the mean

inner potential contribution of the spearhead (hard to prove without access to the raw data). The SD interpretation is mentioned in literature discussions (e.g. Chang et al. 2012), and is clearly at odds with the micromagnetic simulations as well as our now 3D experimental observations (as Reviewer 1 concedes).

8. The change in sign of the normal magnetization in the tip domain is puzzling. What is the physical interpretation? I would have expected a conical spin texture with the direction along the vortex core and/or external field.

We think this is the result of twisting and rotation of the medial domain wall (line 204), and can be reproduced in micromagnetic simulations.

9. How does the reconstruction converge? What metric is used? Ten iterations seem very few.

The 3D vector tomographic reconstruction is based on a gradient descent algorithm as detailed in [Donnelly et al 2018 New J. Phys. 20 083009]. The algorithm initializes with a guessed 3D object (initially zero-filled). A set of projections from this guessed 3D object are generated for each of n measured angles, obtaining a guessed projection dataset $P'_n(x,y)$. The error metric is then calculated by obtaining the difference between the guessed projection dataset, $P'_n(x,y)$ and the experimentally measured projections, $P_n(x,y)$. The error metric is defined as follows:

$$\varepsilon = \sum_n \sum_{x,y} [P'_n(x,y) - P_n(x,y)]^2$$

This error metric is minimized through an iterative process, where the guess of the object is updated at each step by calculating the gradient of the error metric with respect to all three components of the magnetization. The process is repeated in a loop, updating the estimate of the 3D vector object at each iteration until the maximum number of iterations is reached.

The convergence of the 3D reconstruction is shown below, demonstrating that 10 iterations is sufficient.

We have added some more details on this to the methods section:

Line 559: "The algorithm initializes with a guessed 3D object (initially zero-filled). A set of projections from this guessed 3D object are generated for each measured angle. An error metric is calculated by obtaining the sum squared difference between

the guessed projection dataset and the experimentally measured projections. The error metric is minimised by updating the object at each iteration, guided by the gradient of the error metric with respect to all three components of the magnetization. The algorithm converged after 10 iterations.”

10. I am a bit confused by the authors’ statement “Although this mesh resolution is a factor of two greater than the exchange length of magnetite, the choice was necessary to keep the generated mesh files and simulation times within practical limits, and is typically considered an acceptable compromise.” The mesh resolution is excessively large and would be unacceptable to the magnetism community where the mesh size should be half the exchange length, not twice its value.

For finite element simulations (as opposed to finite difference) using MERRILL the recommendation and standard practice in our community is to specify the tetrahedral mesh length to be equal to the exchange length (10 nm) (Ó Conbhúí, P., Williams, W., Fabian, K., Ridley, P., Nagy, L., Muxworthy, A.R., 2018. MERRILL: Micromagnetic Earth Related Robust Interpreted Language Laboratory. *Geochemistry, Geophysics, Geosystems* 19, 1080–1106. <https://doi.org/10.1002/2017GC007279>). Unfortunately that leads to a mesh file size that is prohibitively large (2GB) for the computing resources we had available (simulation would crash due to lack of memory). 5 nm would be completely out of the question! Reducing the mesh resolution to 20 nm was necessary to enable the large number of simulations that needed to be performed. However, following the reviewers comment we did perform one simulation at full 10 nm resolution using an HPCF to confirm that we obtain similar results:

We have added a statement to this effect to the methods and changed “acceptable” to “necessary given the available computing resources”:

Line 582: “Although this mesh resolution is a factor of two greater than the exchange length of magnetite, the choice was a necessary compromise to keep the generated mesh files and simulation times within practical limits given the available computing resources. One simulation at full 10 nm resolution was performed on a high-performance computing facility to confirm that the results are indeed similar.”

11. Please elaborate on the “The methods developed here”. To my best knowledge, the presented methods were simply applied not developed.

True - changed to applied.

12. I suggest rephrasing “oriented predominantly normal to z, i.e., parallel to the basal plane of the particle and going into and out of the plane of the diagram in each half.” As “predominantly lay in the plane perpendicular to the symmetry axis of the spearhead” or something similar.

OK done.

Giant magnetofossil revision review response

We would like to thank all three reviewers for their positive response to our revised manuscript.

Our responses to reviewers' comments are given in red below.

Specific changes made to the manuscript are quoted in blue (Line numbers refer to the tracked changes word document).

Reviewer 1

1. I am happy with the responses and revisions.
One minor fix - as far as I know, the chiton magnetite is still single crystals, not polycrystalline.

We have re-checked the cited reference (Wang, T. *et al.* Mesocrystalline Ordering and Phase Transformation of Iron Oxide Biominerals in the Ultrahard Teeth of *Cryptochiton stelleri*. *Small Struct.* **3**, 2100202 (2022) which confirms that the teeth are NOT single crystals, but instead have a mesocrystalline structure (**Mesocrystalline** refers to mesoscopically structured crystals that are formed through the ordered alignment of nanocrystals, and are somewhat intermediate between single crystal and polycrystal). We have therefore changed the word polycrystalline to mesocrystalline, consistent with the title of the cited work.

Line 77: Mesocrystalline magnetite is also known to be biomineralised in the major lateral teeth of chitons^{19,20}, where it is used as a hardening agent to scrape endolithic algae from rocks.

Reviewer 2

1. I remain concerned about the reproducibility of the work on other magnetofossils and by other researchers. However, as applied here, the findings from this method have direct implications for the magnetofossil community. The manuscript looks to be in good shape after this revision so I do not have additional major comments.

We remain confident in the reproducibility of the methods, based on our extensive follow up work. We hope to publish this soon, providing additional reassurance.

Reviewer 3

1. The authors' response and revision are satisfactory to warrant publication in CEE.

I would suggest adding a quantitative comparison for the micromagnetic simulations using 20 and 10 nm mesh size to the SI. This should include energy contributions and line profiles.

We have added the requested comparison of 20 nm versus 10 nm simulations to the Supplemental, taken along two different profiles, and included a quantitative comparison of the moment and energy densities of the two.

Figure S7. Comparison of micromagnetic simulations performed using a 20 nm mesh (a, c) versus a 10 nm mesh (b, d). Simulations were performed using the same elliptical cross section model used for Fig. 3 and with the [113] axis parallel to the length of the particle (y-z plane). Colours represent the M_x component of magnetisation in all panels. Black line in a and b shows the clipping plane normal to z used to create the images in c and d. Black line in c and d shows the clipping plane normal to x used to create the images in a and b. Results are identical in terms of general features but there are some small differences in detail. The remanent moments are $1 \times 10^{-14} \text{ Am}^2$ for the 20 nm simulation and $0.68 \times 10^{-14} \text{ Am}^2$ for the 10 nm simulation. Total energy densities are -2911 J/m^3 for the 20 nm simulation and -2457 J/m^3 for the 10 nm simulation.

In addition to the changes requested by reviewers 1 and 3, we have taken full note of the editorial requests and made the appropriate changes throughout that can be seen in the tracked changes version of the updated manuscript.

Cheers

Rich